# Structural dynamics of a metal–organic framework induced by $CO_2$ migration in its non-uniform porous structure

Pu Zhao [1,2], Hong Fang [3], Sanghamitra Mukhopadhyay [4], Aurelia Li[5], Svemir Rudić[4], Ian J. McPherson[2,7], Chiu C. Tang[6], David Fairen-Jimenez [5], S.C. Edman Tsang[2] & Simon A.T. Redfern [1]

Stimuli-responsive behaviors of flexible metal–organic frameworks (MOFs) make these materials promising in a wide variety of applications such as gas separation, drug delivery, and molecular sensing. Considerable efforts have been made over the last decade to understand the structural changes of flexible MOFs in response to external stimuli. Uniform pore deformation has been used as the general description. However, recent advances in synthesizing MOFs with non-uniform porous structures, i.e. with multiple types of pores which vary in size, shape, and environment, challenge the adequacy of this description. Here, we demonstrate that the $CO_2$-adsorption-stimulated structural change of a flexible MOF, ZIF-7, is induced by $CO_2$ migration in its non-uniform porous structure rather than by the proactive opening of one type of its guest-hosting pores. Structural dynamics induced by guest migration in non-uniform porous structures is rare among the enormous number of MOFs discovered and detailed characterization is very limited in the literature. The concept presented in this work provides new insights into MOF flexibility.

[1] Department of Earth Sciences, University of Cambridge, Cambridge CB2 3EQ, UK. [2] Department of Chemistry, University of Oxford, Oxford OX1 3QR, UK. [3] Department of Physics, Virginia Commonwealth University, Richmond, VA 23284, USA. [4] ISIS Neutron and Muon Source, Rutherford Appleton Laboratory, Didcot OX11 0QX, UK. [5] Department of Chemical Engineering and Biotechnology, University of Cambridge, Cambridge CB3 0AS, UK. [6] Diamond Light Source Ltd., Harwell Science and Innovation Campus, Didcot OX11 0DE, UK. [7] Present address: Department of Chemistry, University of Warwick, Coventry CV4 7AL, UK. Correspondence and requests for materials should be addressed to P.Z. (email: pu.zhao@chem.ox.ac.uk) or to S.A.T.R. (email: satr@cam.ac.uk)

Flexible metal–organic frameworks (MOFs) are attracting an ever-increasing amount of attention because their stimuli-responsive structures make them highly competitive in various applications such as gas separation[1,2], drug delivery[3], and molecular sensing[4,5]. In particular, some flexible MOFs show stepped gas adsorption isotherms that arise directly from their structural properties. Consequently, gas separation in these MOFs, which is based on the adsorption discrepancy of different gases, can be achieved simply with a small temperature or pressure swing[6]. This greatly reduces the regeneration energies needed and makes flexible MOFs promising in replacing traditional adsorbents for energy-efficient gas separation. Many pioneering works have been carried out to understand MOF structural flexibility related to such stepped gas adsorption isotherms[7–9]. Established theories envisage that when interacting with gas molecules, MOF pores deform for adsorption through linker rotation and/or framework distortion, which leads to the abrupt increase/decrease in gas adsorption capacities[8]. In most cases, it is considered that in a flexible MOF, all pores capable of gas adsorption undergo a similar structural change. This assumption is used by default because most of the flexible MOFs studied in detail have uniform porous structures with only one type of pores that have the same size, shape, and environment[7–9]. For example, MIL-53[10], one of the most studied flexible MOFs with uniform porous structures, shows only one type of quadrilateral prism pores of 8.5 Å diameter[11]. In recent years, MOFs with non-uniform porous structures, i.e., with multiple types of pores which vary in size, shape, and environment, have been shown to also have structural flexibility and unique gas adsorption features[12]. It is thus not appropriate to explain the structural flexibility of these MOFs as a result of the dynamic behavior of one type of pores. One good example is DUT-49, which shows unusual structural flexibility related to stepped negative gas adsorption features[12]. Its negative gas adsorption takes place when gas molecules such as $CH_4$ migrate from its micropores to mesopores, provoking an enormous structural contraction that reduces the amount of gas adsorbed. Noticeably, although this work indicates the internal shuffling of gas molecules between different pores of DUT-49, this claim was unfortunately not supported by any dynamic study. It motivated us to perform a detailed dynamic analysis of guest molecules inside a flexible non-uniform porous MOF. We believe that this well addresses a fundamentally important issue in the understanding of structural transitions and guest-host interactions in flexible MOFs.

Here, we decide to study zeolitic imidazolate framework 7 (ZIF-7) which has a non-uniform porous structure showing structural flexibility and stepped adsorption isotherms during the uptake of guest molecules such as $CO_2$[13]. It is an important member of the MOF subgroup, ZIFs, some of which are noted for their high hydrothermal stability and structural similarity to aluminosilicate zeolites[14]. ZIF-7 ($Zn(bIm)_2$, bIm: benzimidazolate) is composed of zinc atoms connected by bIm linkers in tetrahedral coordination. As indicated by the pore size distribution analysis (Supplementary Fig. 1 and Supplementary Table 1), ZIF-7 has four types of pores, including two types of six-member-ring pores, one type of four-member-ring pores, and the sodalite cage (Fig. 1a). Type A six-member-ring pore (pore A) has the largest void space in ZIF-7 and has long been considered the one to proactively open for gas adsorption[15–17]. However, we found that the structural flexibility and stepped $CO_2$ adsorption isotherms of ZIF-7 is in fact related to $CO_2$ migration between its different types of pores, something that we implied in our previous work[18]. Although formerly we were prohibited from giving a clear description of the migration route due to the lack of evidence, in this present work we are able to include extensive in situ experimental and computational analysis revealing the $CO_2$ migration pathways and its relationship with the adsorption-stimulated structural transitions in ZIF-7. It is demonstrated that in fact $CO_2$ molecules first weakly bind to the adjacent type B six-member-ring pore (pore B) when pore A is closed. With the accumulation of $CO_2$ molecules in pore B, the shared linker between pores A and B rotates in a manner that leads to $CO_2$ flowing from pore B into pore A, resulting in an abrupt increase in $CO_2$ adsorption capacity and inducing an overall structural transformation in ZIF-7. Our results explain the long-discussed nature of ZIF-7 structural flexibility and its $CO_2$ adsorption mechanism. We believe that this sequential pore filling mechanism discussed here will contribute to the fundamental understanding of the structural−property relationships in flexible MOFs.

## Results

**Initial $CO_2$ adsorption.** Figure 1b shows the $CO_2$ adsorption isotherms of ZIF-7 at 195 and 298 K (see Supplementary Table 2 for the tabulated data). At 298 K, a $CO_2$ adsorption transition pressure is observed at 40−60 kPa. Below 40 kPa, ZIF-7 is not able to adsorb much $CO_2$ (0.2 mmol g$^{-1}$). At 60 kPa, ZIF-7 reaches a much higher $CO_2$ adsorption capacity (1.3 mmol g$^{-1}$). The structural transformation corresponding to this abrupt increase in $CO_2$ adsorption capacity has been reported in our previous work (ZIF-7-II to ZIF-7-I phase transition, Supplementary Fig. 2 and Supplementary Table 3)[19]. As seen in Fig. 1c, the six-member-ring pore A shows a structural change induced by linker rotation, which results in an increase in pore size for $CO_2$ accommodation (Supplementary Fig. 1 and Supplementary Table 1). The dynamic behavior of the linkers has previously been described as an opening of the gate of pore A. However, if we look closely at the structure of pore A at high $CO_2$ pressure (ZIF-7-I phase), it can be noticed that the window (or gate) of pore A, whose diameter is 3 Å, is still not large enough for easy access of $CO_2$ molecules whose kinetic diameter is 3.3 Å[20]. Thus, $CO_2$ adsorption in ZIF-7 is not simply a consequence of pore A window expansion.

$CO_2$ adsorption at 195 K offers additional information that helps further understand the process in the structure. At this temperature, $CO_2$ adsorption takes place in two stages. At low $CO_2$ pressure (10 kPa), ZIF-7 reaches two-thirds of its full $CO_2$ adsorption capacity at current temperature (2.7 mmol g$^{-1}$). Further $CO_2$ adsorption takes place over a narrow pressure range before its full $CO_2$ adsorption capacity is reached at 30 kPa (4.2 mmol g$^{-1}$). In situ synchrotron X-ray powder diffraction (SXRD) was employed to monitor $CO_2$ adsorption in ZIF-7 at 195 K (Fig. 2a), in the hope of relating both $CO_2$ adsorption stages with ZIF-7 structural transformations, as reported previously[13,21]. It is found that the ZIF-7-II to ZIF-7-I phase transition takes place over a higher pressure range and is related to the second $CO_2$ adsorption stage. The amount of $CO_2$ adsorbed during the ZIF-7-II to ZIF-7-I phase transition at 195 K (1.5 mmol g$^{-1}$) is comparable to that at 298 K (1.1 mmol g$^{-1}$). No structural transformation can be observed to rationalize the first $CO_2$ adsorption stage at 195 K. This means that ZIF-7 was in ZIF-7-II phase when the initial $CO_2$ adsorption took place. As pore A in ZIF-7-II phase has no $CO_2$ adsorption capability due to its narrow pore space, it is reasonable to conclude that the initial $CO_2$ adsorption at 195 K takes place somewhere other than pore A. Moreover, initial $CO_2$ adsorption appears to lead to the ZIF-7-II to ZIF-7-I phase transition in which the second $CO_2$ adsorption stage occurs.

Previously, the six-member-ring pore B adjacent to pore A was identified as the preferred $CO_2$ adsorption site (Fig. 1a) when we studied $CO_2$ adsorption geometry in ZIF-7-I at 300 K by neutron

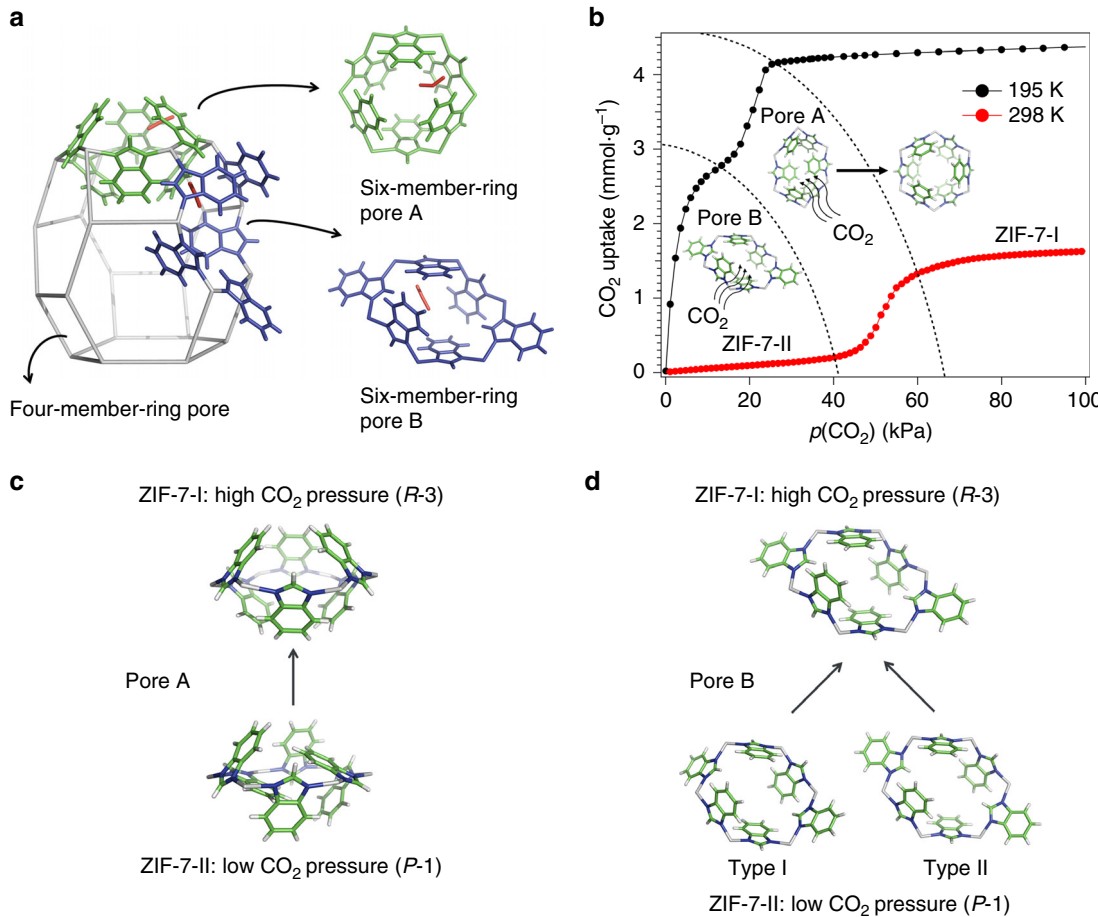

**Fig. 1** ZIF-7 structure and its $CO_2$ adsorption behavior. **a** The building unit, sodalite cage, of ZIF-7 with two types of six-member-ring pores (A, B) and one type of four-member-ring pores on its walls. Part of the framework is simplified by replacing Zn−bIm−Zn with Zn−Zn (bIm = benzimidazolate, $C_7N_2H_5$, Zn: gray). In our previous work[18], we have demonstrated $CO_2$ molecules prefer to be adsorbed in six-member-ring pores, as shown here. In addition, there are more $CO_2$ molecules adsorbed in pore B than in pore A. $CO_2$: red. The symmetry of $CO_2$ has been disregarded for clarity. **b** $CO_2$ adsorption isotherms of ZIF-7 at 195 and 298 K, $p_{CO2} = 1$–100 kPa, illustrated by the structural behaviors of ZIF-7. **c** The structural change of pore A in the ZIF-7-II to ZIF-7-I phase transition. Zn: gray; N: blue; C: green; H: white. **d** The structural change of pore B in the ZIF-7-II to ZIF-7-I phase transition. Zn: gray; N: blue; C: green; H: white. See the crystal structures of ZIF-7-I and ZIF-7-II in Supplementary Fig. 2 and Supplementary Table 3

powder diffraction (ND)[18]. Pore B was later demonstrated to be a preferred adsorption site for $CO_2$ and some other guest molecules by works from others[13,21,22]. Since pore B has identical structures in both ZIF-7-I and ZIF-7-II phases (Fig. 1d), it is possible that pore B in ZIF-7-II is where the $CO_2$ adsorption at 195 K begins. However, it is difficult to validate this hypothesis by directly determining $CO_2$ locations in ZIF-7-II at 195 K using diffraction methods. This is because ZIF-7-II has low symmetry (P-1) and a large unit cell (V = 7917 Å³), Rietveld refinement based on diffraction data is not able to define a reliable ZIF-7-II ($CO_2$) crystal structure. Hence, we needed to find alternative evidence in order to identify the location at which $CO_2$ adsorption in ZIF-7-II begins. Since transport diffusion can reflect the collective motion of gas molecules caused by local concentration gradients, $CO_2$ transport diffusivity in ZIF-7-I was measured to show that, to access the structure, $CO_2$ adsorbs preferentially in pore B rather than in pore A. The measurement was done using quasi-elastic neutron scattering (QENS) at 298 K and a $CO_2$ loading of 1.3 mmol g⁻¹. The length and time scales of QENS match those of molecular dynamics (MD) simulations and good agreement are often found between experimental and theoretical results. Sample dependence, due to factors such as surface barriers and structural defects, can be eliminated in QENS results. Detailed QENS data analysis can be found in Supplementary Discussion

(Supplementary Figs. 3–5). The measured $CO_2$ transport diffusivity in ZIF-7-I ($6.3(7) \times 10^{-9}$ m² s⁻¹, Fig. 2b) was compared with those experimentally obtained in ZIF-8 at the same temperature ($0.4(3) \times 10^{-9}$ m² s⁻¹)[23–26]. We did this comparison because ZIF-8 is an analog of ZIF-7 but has only one type of $CO_2$ hosting pore (see pore size distribution analysis in Supplementary Fig. 1 and Supplementary Table 1) connected by six-member-ring windows with similar size (diameter 3.4 – 4.1 Å) and geometry to those of pore A in ZIF-7-I (Supplementary Fig. 6)[14,23]. In contrast, pore B in ZIF-7-I and ZIF-7-II has larger windows of 4.6 – 5.3 Å diameter. The one-order-of-magnitude larger $CO_2$ transport diffusivity in ZIF-7-I compared with ZIF-8 indicates that, in accessing ZIF-7-I, $CO_2$ prefers pore B to pore A. Since the structure of pore B is identical in both ZIF-7-I and ZIF-7-II phases and pore A has smaller pore space in ZIF-7-II than in ZIF-7-I, pore B is expected to be the initial site for $CO_2$ adsorption in ZIF-7-II.

The absence of initial $CO_2$ adsorption prior to the ZIF-7-II to ZIF-7-I phase transition at 298 K (Fig. 1b) can be explained by examining the crystal structure of $CO_2$-adsorbed ZIF-7-I determined by ND under similar conditions[18] (Fig. 3a). It can be found that once adsorbed, $CO_2$ molecules bind to pore B through weak hydrogen bonding ($O_{CO2}\cdots H_B = 2.64(2)$ Å). Thus it is likely that $CO_2$ only gets adsorbed in pore B at low temperature

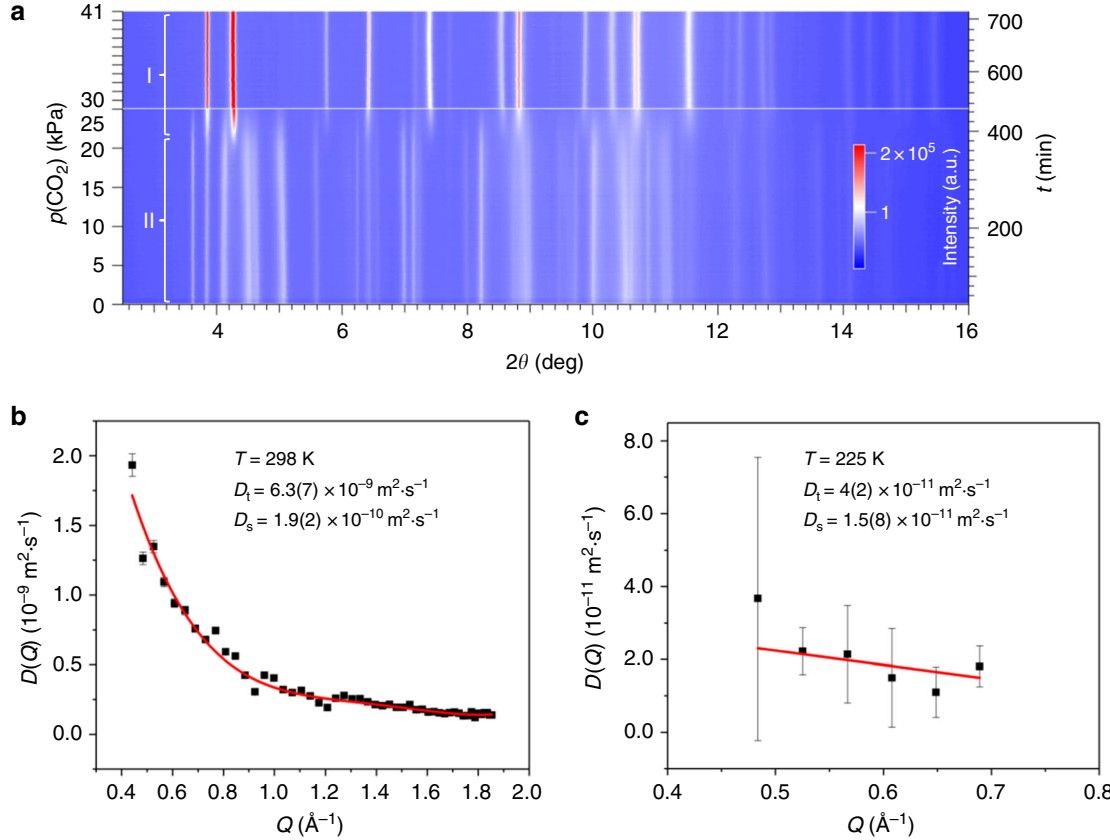

**Fig. 2** Phase transition and $CO_2$ diffusion in ZIF-7. **a** In situ SXRD data of ZIF-7 at $T = 195$ K, $p(CO_2) = 0$–41 kPa. $\lambda = 0.82634(1)$ Å. At 0–23 kPa, the diffraction pattern corresponds to ZIF-7-II. Above 30 kPa, the diffraction pattern of ZIF-7-I dominates. **b** The $Q$ dependence of the $CO_2$ transport diffusion coefficient $D$ in ZIF-7 at a $CO_2$ loading of 1.3 mmol g$^{-1}$ and $T = 298$ K, fitted by a 4th-order polynomial function. Error bars are calculated using Supplementary Equation 6. $D_t$: transport diffusivity. $D_t = D(0)$. $D_s$: self-diffusivity. $D_t = D_s(\partial \ln p / \partial \ln c)$. $p$ is $CO_2$ pressure in kPa and $c$ is $CO_2$ uptake at $p$ in mmol g$^{-1}$. See Supplementary Discussion for detailed analysis. **c** The $Q$ dependence of the $CO_2$ transport diffusion coefficient $D$ in ZIF-7 at a $CO_2$ loading of 1.3 mmol g$^{-1}$ and $T = 225$ K, fitted by a linear function. Error bars are calculated using Supplementary Equation 6

when the thermal motion of $CO_2$ is significantly minimized. This conclusion is validated by comparing the $CO_2$ self-diffusivity in ZIF-7 at 225 and 298 K. We used 225 K instead of 195 K in the measurement because of the detection limit. Since 195 K is the sublimation point of $CO_2$ at 101 kPa, at this temperature $CO_2$ diffusion is too slow to be measured accurately. Self-diffusion describes the elementary jumps of an individual gas molecule between adsorption sites. The low value of $CO_2$ self-diffusivity measured at 225 K ($1.5(8) \times 10^{-11}$ m$^2$ s$^{-1}$, Fig. 2c) shows that $CO_2$ molecules exhibit negligible jumping between adsorption sites at or below this temperature. This facilitates $CO_2$ storage into pore B, which corresponds to the initial $CO_2$ adsorption prior to the ZIF-7-II to ZIF-7-I phase transition at 195 K. In contrast, $CO_2$ self-diffusivity measured at 298 K ($1.9(2) \times 10^{-10}$ m$^2$ s$^{-1}$, Fig. 2b) shows that $CO_2$ exhibits rapid jumping between adsorption sites at this temperature, which indicates that $CO_2$ has enough energy to overcome the stabilization provided by the weak hydrogen bonds between $CO_2$ and pore B, leading to the absence of initial $CO_2$ adsorption.

In order to corroborate the above hypothesis, grand canonical Monte Carlo (GCMC) molecular simulations were performed using ZIF-7-I and ZIF-7-II structures separately. A similar strategy was followed in the past on ZIF-7[13] and ZIF-8[27,28]. Figure 4a shows the comparison of the experimental and simulated $CO_2$ isotherms of ZIF-7. In the simulated isotherms, we highlighted those points (solid symbols) corresponding to the phases observed experimentally. At 298 K, the simulated isotherms match nicely the experimental data, with a slight

overprediction at lower pressures for ZIF-7-II. At 195 K, the simulated isotherm for ZIF-7-II—with blocked pores A—matches the experimental isotherm very well. Figure 4b shows the snapshot and density distribution of adsorbed $CO_2$ molecules in ZIF-7-II at 10 kPa and 195 K. These results confirm the initial $CO_2$ adsorption in pore B of ZIF-7-II. Although $CO_2$ uptake in ZIF-7-I after the phase transition is underpredicted by the GCMC simulations, the results match those from the previous simulations on ZIF-7[13]. As indicated previously, more efficient packing of $CO_2$ molecules may take place in ZIF-7-I at 195 K, which might be the reason for the underprediction by our GCMC simulations.

**$CO_2$ migration and ZIF-7 structural dynamics.** At this point, it is clear that $CO_2$ adsorption initially takes place in pore B. However, how $CO_2$ migrates from pore B to pore A and what induces the ZIF-7-II to ZIF-7-I phase transition remains uncertain. The crystal structure of $CO_2$-adsorbed ZIF-7-I determined by ND at 300 K is used to examine the interactions between $CO_2$ molecules and both pores as well as the spatial relationship between them. Pores A and B share a common bIm linker and $CO_2$ molecules adsorbed in both pores are stabilized by hydrogen bonding with this linker (Fig. 3a). This common linker is named the star linker for the convenience of explanation. The $CO_2$ molecule in pore B has one hydrogen bond with the imidazole ring (Im) of the star linker. At the initial $CO_2$ adsorption stage, as the number of $CO_2$ molecules adsorbed in pore B increases,

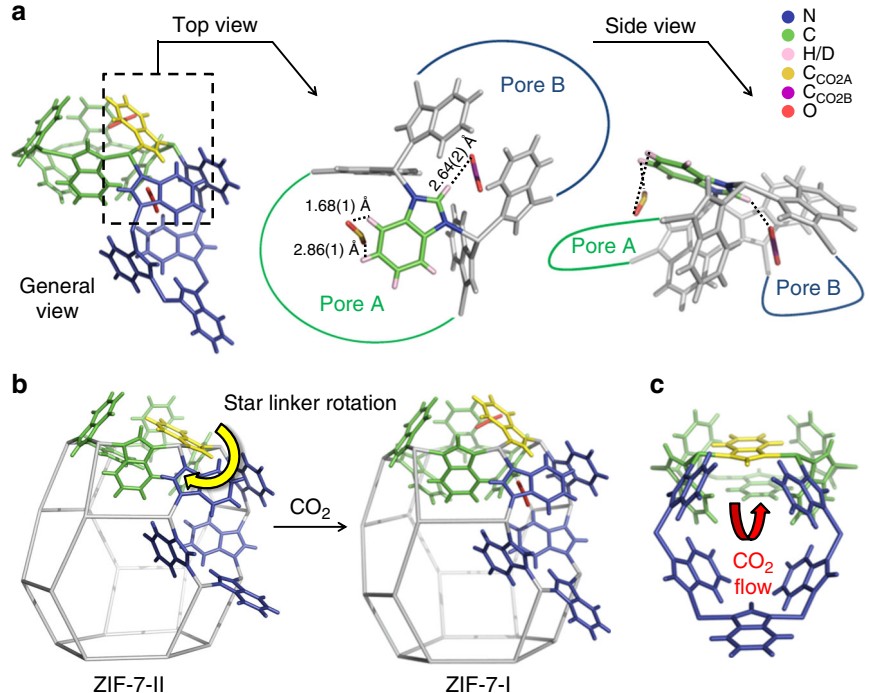

**Fig. 3** $CO_2$ migration route and ZIF-7 structural dynamics. **a** Spatial relationship between pores A and B as well as adsorbed $CO_2$, determined by ND. In general view, pore A: green, pore B: blue, star linker: yellow, $CO_2$: red. The symmetry of $CO_2$ has been disregarded for clarity. In top and side view, star linker: N, blue; C, green; H/D, pink. C of $CO_2$ in pore A: yellow, C of $CO_2$ in pore B: purple, O: red. **b** Star linker rotation caused by $CO_2$ adsorption in pore B, leading to the ZIF-7-II to ZIF-7-I phase transition. **c** The channel between pores A and B in ZIF-7-II, allowing $CO_2$ to migrate from pore B to A. pore A: green, pore B: blue, star linker: yellow

the hydrogen bond between $CO_2$ and Im makes the star linker rotate in a way that eventually induces the ZIF-7-II to ZIF-7-I phase transition (Fig. 3b). Meanwhile, the migration of $CO_2$ from pore B to pore A happens fast via the channel between two pores (Fig. 3c). The $CO_2$ molecule is stabilized in pore A by two hydrogen bonds with the benzene ring (Bz) of the star linker. These two hydrogen bonds are stronger than ($O_{CO2}\cdots H_{Bz}$ = 1.68(1) Å) or comparable with ($O_{CO2}\cdots H_{Bz}$ = 2.86(1) Å) that between $CO_2$ and Im in pore B ($O_{CO2}\cdots H_{Im}$ = 2.64(2) Å). Due to their spatial relationships, the hydrogen bond formed between $CO_2$ and Bz in pore A inhibits star linker rotation caused by the hydrogen bond between $CO_2$ and Im in pore B (Fig. 3a). Thus at this point, the ZIF-7-II to ZIF-7-I phase transition is completed. It is worth mentioning that, although there are two types of pore B in ZIF-7-II (Fig. 1d), type I pore B should be the one that is responsible for the initial $CO_2$ adsorption. This is because its shared linker with pore A has the most distinctive rotation during the ZIF-7-II to ZIF-7-I phase transition, and is associated with the highest activation barrier for the phase transition. This is further confirmed by our inelastic neutron scattering (INS) study of the changes in the vibrational behavior of the linkers during $CO_2$ adsorption. INS is a highly quantitative spectroscopic technique whose results are generally comparable to that of theoretical calculation. See Supplementary Discussion for detailed analysis (Supplementary Fig. 7 and Supplementary Table 4).

In order to further validate the $CO_2$ migration route concluded from experimental data, we complemented our study with ab initio calculations based on density functional theory (DFT). Energetics were studied when $CO_2$ molecules are adsorbed at different locations, including in pore B or in the channel between pores A and B before the ZIF-7-II to ZIF-7-I phase transition, as well as in pore A or in both A and B pores after phase transition

(see Supplementary Fig. 8 for models). In each case, $CO_2$ molecules are localized near the bIm linkers after energy relaxation. The averaged adsorption energy of each $CO_2$ molecule ($\Delta E_{CO2}$) is given by

$$\Delta E_{CO2} = -\frac{E - E_f - N_{CO2} \times E_{CO2}}{N_{CO2}} \qquad (1)$$

where $E$, $E_f$, $E_{CO2}$, and $N_{CO2}$ are the total energy of the system, the framework energy, the energy of a single $CO_2$ molecule, and the number of $CO_2$ molecules, respectively. Before phase transition, the $CO_2$ adsorption energy in pore B is only 6.9 kJ mol$^{-1}$ per molecule larger than that in the channel between pores A and B. Such a small energy difference suggests that $CO_2$ molecules are likely to stay in pore B initially but can easily move into the channel under thermal excitation. After phase transition, the $CO_2$ adsorption energy in both A and B pores is 14.1 kJ mol$^{-1}$ per molecule larger than that in pore A. This suggests that $CO_2$ molecules tend to first fill pore B before going into pore A. The combined picture of these results is that in the adsorption process: $CO_2$ molecules migrate from pore B to pore A via the open channel. This is in line with the conclusions we derived from experimental results.

It is noted that DFT calculations were carried out at 0 K, thus temperature effect (e.g., atomic vibrations) was not taken into account. GCMC simulations allow us to have a complete thermodynamic description of the $CO_2$ interactions with ZIF-7 under experimental conditions. DFT calculations estimate that the energy difference between ZIF-7-I and ZIF-7-II framework ($\Delta E_f$) is around 29.9 kJ mol$^{-1}$ (one unit cell). Upon adsorption, this energy difference is expected to be overcome by the affinity of $CO_2$. Figure 4c shows the heat of adsorption ($Q_{st}$) as a function of $CO_2$ uptake, obtained from GCMC simulations. The difference

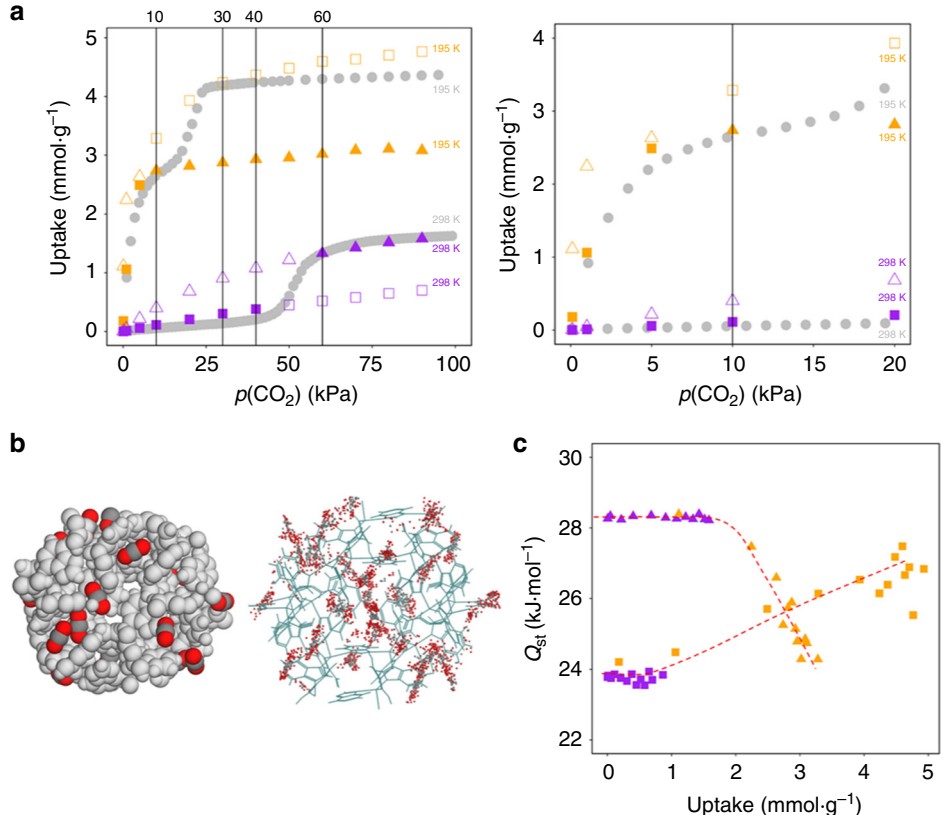

**Fig. 4** GCMC simulations. **a** $CO_2$ adsorption isotherms of ZIF-7. Left: a comparison of experimental (gray circles) and simulated adsorption isotherms, using ZIF-7-I (triangles) and ZIF-7-II (squares) structures at 195 (orange) and 298 K (purple). Right: a detailed comparison at low pressure. Solid triangles and squares in the simulated isotherms correspond to the phases observed in the experimental isotherms. The vertical lines highlight the range of pressures at which the phase transitions occur: 10–30 kPa at 195 K and 40–60 kPa at 298 K. **b** Snapshot (left) and density distribution (right) of adsorbed $CO_2$ molecules in ZIF-7-II at 10 kPa and 195 K. **c** Heat of adsorption of $CO_2$ in ZIF-7-I (triangles) and ZIF-7-II (squares) at 195 (orange) and 298 K (purple). Red dotted lines are included as eye-guides

in the heat of adsorption of $CO_2$ ($\Delta Q_{st}$) in the transition region can be given by[12]

$$\Delta Q_{st} = N'_{CO2} \times (Q_{st}\text{-I} - Q_{st}\text{-II}) \qquad (2)$$

where $N'_{CO2}$ is the number of $CO_2$ molecules adsorbed in one ZIF-7 unit cell, $Q_{st}$-I and $Q_{st}$-II are the $Q_{st}$ obtained for ZIF-7-I and ZIF-7-II, respectively. At 298 K, $\Delta Q_{st} = 30.1$ kJ mol$^{-1}$ at the uptake of 1.3 mmol g$^{-1}$ ($N_{CO2} = 7$) where the ZIF-7-II to ZIF-7-I phase transition completes. This value is comparable to $\Delta E_f$. At 195 K, maximum $\Delta Q_{st}$ is 34.5 kJ mol$^{-1}$ at the uptake of 2 mmol g$^{-1}$ ($N_{CO2} = 11$), also comparable to $\Delta E_f$. These results underpin that initial adsorption is needed to compensate for the energy penalty of the structural transition. However, as the energy barrier of the structural transition is difficult to be estimated and $CO_2$ uptake in ZIF-7-I at 195 K after the transition is underpredicted by the GCMC simulations, the simulated amount of $CO_2$ needed to trigger the phase transition slightly deviates from the experimental value.

## Discussion
In conclusion, a refined ZIF-7 $CO_2$ adsorption mechanism is presented in this work. It reveals that the $CO_2$-adsorption-stimulated structural flexibility of ZIF-7 is due to $CO_2$ migration between its different guest-hosting pores upon pressure. This mechanism is extremely uncommon and is distinct from the majority of guest-host interactions reported for other MOFs. Overall, the relationship between guest and host dynamics discussed here provides new insights into MOF flexibility. This work

also offers new methods for analyzing adsorption mechanisms in other known flexible MOFs, particularly those with non-uniform porous structures, by evaluating the adsorption strength in each type of pores and the extent of guest migration with respect to the phase change. This may lead to the selective filling of a specific type of pores without triggering the associated phase transformation, which could result in the rational design of optimized stimuli-responsive behaviors.

## Methods
**Synthesis.** ZIF-7 was synthesized using a solvothermal method. 0.75 g of zinc nitrate hexahydrate (Zn(NO$_3$)$_2$·6H$_2$O, 2.52 mmol) and 0.25 g of benzimidazole (HbIm, C$_7$H$_6$N$_2$, 2.05 mmol) were first dissolved in 75 ml of fresh dimethylformamide (DMF). The resultant solution was then sealed into a 100 ml Teflon-lined stainless steel autoclave. The autoclave was heated at 400 K for 48 h. After naturally cooling to room temperature, white platy crystals were isolated after the mother liquor was removed. The solid product was washed thoroughly with methanol. The average yield was around 0.34 g (97% based on HbIm). All chemicals employed were commercially available (Sigma-Aldrich and Acros Organics), with purity of 98% or above, and were used as received.

**Laboratory characterization.** The crystal structures of ZIF-7 samples were characterized by powder X-ray diffraction (XRD) using a Bruker D8 Advance X-ray diffractometer operated at 40 kV and 40 mA with Cu Kα$_1$ radiation ($\lambda = 1.54056$ Å) (Supplementary Fig. 9). The XRD data of the as-synthesized sample were compared with that calculated from the ZIF-7-I crystallographic model in the literature[14]. ZIF-7-I purity was around 100%. The morphology of ZIF-7 samples was examined by scanning electron microscopy using a JEOL JSM-5510 microscope (Supplementary Figs. 10 and 11). $CO_2$ adsorption isotherms were measured by a Hiden Isochema IGA gravimetric analyzer. The as-synthesized sample was activated at 400 K in air for 48 h before each measurement. Before data collection, the sample was vacuumed

in situ (~$10^{-4}$ mbar) at room temperature for 3 h to remove any remaining trace guest molecules.

**In situ synchrotron X-ray powder diffraction (SXRD).** In situ SXRD data were collected at beamline I11, Diamond Light Source, Harwell, UK. The energy of the incident X-ray was set at ~15 keV[29]. The wavelength ($\lambda = 0.82634(1)$ Å) and the $2\theta$ zero point were determined by fitting the diffraction data of high-quality silicon powder (SRM640c).

The as-synthesized sample was first activated at 400 K in air for 24 h. The activated sample was then loaded into a quartz glass capillary (0.5 mm). Glass wool was packed on top of the sample and the capillary was then fixed onto a custom-made gas cell. Before data collection, the sample cell was evacuated to $10^{-5}$ Pa at room temperature for 3 h to remove any remaining trace guest molecules from the sample. $CO_2$ pressure was controlled by a custom-made gas loading system. After each $CO_2$ loading, 2 min was allowed before data collection for the system to reach equilibrium.

Since sample spinning was prohibited due to the position of the gas cell, the sample was rocked through $\pm 10°$ about the $\omega$-axis to improve powder averaging. SXRD data were collected with Mythen II position sensitive detectors (PSD) over the $2\theta$ range of 2–90°, with two datasets offset by 0.25° ($2\theta$) collected (each for 30 s). These datasets were subsequently automatically merged to remove gaps in the data where the microstrips (X-ray detection modules) join. PSD and data treatment have been described in detail in the literature[30].

**Quasi-elastic neutron scattering (QENS).** QENS data were collected using the IRIS indirect geometry time-of-flight (TOF) spectrometer at ISIS Neutron and Muon Source, Rutherford Appleton Laboratory, Harwell, UK. At IRIS, the sample first got a white beam of neutrons containing a band of energy. After scattering, the crystal analyzer (the (002) plane of pyrolithic graphite) chose a single final energy (1.84 meV) to send to detectors. Scattered neutrons were detected over an angular range of $2\theta = 25$–160°. The instrumental resolution and detector efficiencies were calibrated by fitting the spectrum of a vanadium standard. The elastic energy resolution was 17.5 µeV. The energy transfer range was $-0.4$–0.4 meV and the $Q$ range was 0.42–1.85 Å$^{-1}$. IRIS was also built with long-wavelength diffraction capability. The $d$ range was 1–12 Å with $\Delta d/d = 2.5 \times 10^{-3}$.

The as-synthesized sample was first activated at 400 K in air for 24 h. The activated sample (ca. 4.6 g) was then wrapped into two pieces of aluminum foil to make a sample lining fitted into the annular space of an aluminum cylinder sample cell (Ø 24/28 mm $\times h$ 65.6 mm). The thickness of the sample lining was about 2 mm. Glass wool was placed in the top of the sample cell. The sample cell was then sealed with indium and connected to a custom-made gas loading system. During data collection, the temperature of the sample was controlled by a helium cryostat and thermocouples. QENS spectra were first recorded when the sample was under vacuum. At 15 and 225 K, the accumulated proton current was 600 µA for good statistics. At 298 K, the accumulated proton current was 1146.8 µA. $CO_2$ was then loaded into the sample cell at 298 K to enable ZIF-7 to reach a $CO_2$ uptake of 1.3 mmol g$^{-1}$. For QENS data of the loaded sample collected at 15, 225, and 298 K, the accumulated proton current was 600 µA. Diffraction data were collected simultaneously with QENS data collection. $CO_2$ loading was measured using the method employed in the literature[31–33]. It is also described in detail in Supplementary Methods.

**Inelastic neutron scattering (INS).** INS data were collected using the TOSCA TOF spectrometer at ISIS Neutron and Muon Source, Rutherford Appleton Laboratory, Harwell, UK. TOSCA has a wide energy transfer range of $-2.5$–1000 meV ($-20$–8050 cm$^{-1}$) and a high energy resolution of ca. 1.25% $\Delta E/E$. The sample environment at TOSCA was similar to that at IRIS. Before data collection, the as-synthesized sample was first activated at 400 K in air for 24 h. The activated sample (ca. 4.6 g) was then wrapped into two pieces of aluminum foil to make a sample lining for a stainless steel cylinder sample cell (Ø 16 mm $\times h$ 76 mm). Glass wool was placed in the top of the sample cell to prevent any sample spillage during the experiment. The sample cell was then sealed with a copper O-ring and connected to a custom-made gas loading system. The sample chamber was evacuated to $10^{-5}$ Pa at room temperature for 16 h to remove any remaining trace guest molecules from the sample. During data collection, the temperature of the sample was controlled by a closed cycle refrigerator (CCR) and was kept below 10 K to minimize the thermal motion of the sample and $CO_2$ molecules. INS spectra were first recorded when the sample was under vacuum. The accumulated proton current was 3604 µA for good statistics. $CO_2$ was then loaded into the sample cell at 298 K to enable ZIF-7 to reach a $CO_2$ uptake of 1.3 mmol g$^{-1}$. $CO_2$ loading was measured using the same method as that in the QENS experiment. During data collection of the loaded sample, the accumulated proton current was 3558.6 µA.

**Density functional theory (DFT) calculation.** Models for the DFT calculations were built based on the crystal structures from the literature[18,19]. One ZIF-7 unit cell was used with a chemical formula of $Zn_{18}C_{252}N_{72}H_{180}$. Perdew-Burke-Ernzerhof (PBE) generalized gradient approximation (GGA)[34] implemented in the VASP package[35] was used. The projector augmented wave (PAW)[36]

pseudopotential method was employed. Due to the large unit cell of ZIF-7, only Gamma point was sampled. The cutoff energy was 400 eV. The shape of the unit cell and atomic positions were allowed fully relaxed during the optimization. The energy convergence for the self-consistent electronic relaxation was set to be $10^{-6}$ eV and the force convergence was set to be 0.01 eV Å$^{-1}$. To count in the weak dispersive interaction, van der Waals interactions (as implemented in the DFT-D2 scheme[37,38]) were considered during the calculations. We included a typical input file of our DFT calculations as a supporting file (Supplementary Data 1). Details of the calculation can be found in this file.

**Grand canonical Monte Carlo (GCMC) molecular simulations.** $CO_2$ adsorption isotherms in ZIF-7 were investigated using grand canonical Monte Carlo (GCMC) simulations performed in the multi-purpose code RASPA[39]. ZIF-7-I and ZIF-7-II models were built based on the crystal structures from the literature[18,19]. Framework atoms were kept fixed at the crystallographic positions. In order to correctly describe the absence of $CO_2$ molecules in pores A before the ZIF-7-II to ZIF-7-I phase transition, pores A were all blocked in ZIF-7-II during the simulation. We used the standard Lennard-Jones (LJ) 12-6 potential to model the interactions between the framework and fluid atoms. In addition, a Coulomb potential was used for fluid-fluid interactions. The parameters for framework atoms were derived from the Universal Force Field[40] and those previously developed for ZIF-8[27]. $CO_2$ molecules were modeled using the TraPPE potential with charges placed on each atom and at the center of mass (Supplementary Table 5)[41]. EQeq was used to assign the partial charges of the framework. The Lorentz-Berthelot mixing rules were employed to calculate fluid-solid LJ parameters, and LJ interactions beyond 12.8 Å were neglected. The Ewald summations method was used to compute the electrostatic interactions. Up to 50,000 Monte Carlo cycles were performed, the first 50% of which were used for equilibration, and the remaining steps were used to calculate the ensemble averages. Monte Carlo moves consisted of insertions, deletions, displacements, and rotations. In a cycle, $N$ Monte Carlo moves are attempted, where $N$ is defined as the maximum of 20 or the number of adsorbates in the system. To calculate the gas-phase fugacity we used the Peng-Robinson equation of state[42]. The isosteric heat of adsorption ($Q_{st}$) was calculated using the fluctuation theory[43]. Input files are included in Supplementary Data 2.

## Data availability

The data that support the findings of this study are available from the authors on reasonable request. See author contributions for specific datasets.

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

## Acknowledgements

This work was supported by the Cambridge Commonwealth, European and International Trust, and UK Science and Technology Facilities Council. The authors thank Diamond Light Source (UK) for the SXRD facility and ISIS Neutron and Muon Source (UK) for QENS and INS facilities. Dr. James Taylor from ISIS Neutron and Muon Source is thanked for his assistance in $CO_2$ adsorption isotherm measurements. Dr. Giulio I. Lampronti from the University of Cambridge is acknowledged for discussions.

## Author contributions

P.Z. carried out material synthesis and laboratory characterization. In situ SXRD was performed by P.Z. and C.C.T. QENS data were collected by P.Z., S.M., and S.A.T.R. INS data were collected by P.Z., and S.R. H.F. performed the DFT calculations. A.L. carried out GCMC simulations, supervised by D.F.-J. The manuscript was written and revised by P.Z. H.F., I.J.M., S.C.E.T., and S.A.T.R. The project was planned and directed by P.Z. and supervised by S.A.T.R. All authors discussed the results and commented on the manuscript.
