## [Peer Review File · Nature Communications]

Reviewers' comments:

Reviewer #1 (Remarks to the Author):

In their work "Structural flexibility of a metal-organic framework induced by CO₂ migration in its non-uniform porous structure" Zhao et al. address the structural transition and host-guest interactions in the flexible metal-organic framework (MOF) ZIF-7 by extensive in situ experimental efforts aided by computational analysis. In contrast to most existing "static" studies on flexibility in MOFs, some of them cited herein, this work addresses the adsorption mechanism and flexibility in ZIF-7 by experimental diffusion analysis of the guest, in this case carbon dioxide, which makes it fundamentally interesting for the growing field of stimuli-responsive MOFs. In their findings, the authors refine the adsorption mechanism in ZIF-7 with findings contrary to previous investigations (in part by the authors themselves). In my opinion, the presented work in its current form does not allow the reader to fully understand the fundamental differences to the previously proposed mechanism.

In the introduction line 38 the authors state:

"Established theories envisage that when interacting with gas molecules, MOF pores open for adsorption through linker rotation and/or framework distortion."

This is only true for a portion of flexible MOFs generally referred to as gate-opening MOFs. Other materials, such as the cited DUT-49 or MIL-53, exhibit contraction of the pores upon interaction with gas molecules. The authors should refine their statement in this regard.

Furthermore, the authors refer to the pressure at which a structural transition occurs as: "adsorption critical pressure". This might be mistaken with the critical pressure of the gas. I suggest changing the applied terminology to transition pressure instead.

In regards to the title, the authors state that (line 43):

"In general, it is considered that in a flexible MOF, all pores capable of gas adsorption behave in a similar way. This assumption is used by default because most of the flexible MOFs studied have uniform porous structures."

I feel that in this context the terminology uniform, and later on non-uniform porous structure, might not be an adequate description and potentially misleading for readers. To avoid confusion, the authors should provide detailed example of what they understand to be uniform and non-uniform pore structures instead of just citing three recent reviews about flexible MOFs in general (reference 7-9) to support their definition.

The authors describe DUT-49, a MOF that exhibits a pore structure consisting of micro and mesopores, as non-uniform porous structure and set the investigated ZIF-7 in context to DUT-49. Thus, a MOF with channeled pore structure such as MIL-53 exhibits a uniform pore structure? Because this terminology is used in the title, I feel a more detailed definition would support the readability of the manuscript.

To further define a non-uniform pore structure, the authors can present pore size distribution analysis of ZIF-8. In fact, I believe it would strengthen the proposed mechanism if an evolution of the pore size distribution in ZIF-7-I and ZIF-7-II would be provided and set in context to the investigated adsorption sites. In this context, the authors state:

"There are four types of pores in ZIF-7, including two types of six-member-ring pores, one type of four-member-ring pores, and the Sodalite cage." In previous literature on ZIF-7 these pores are usually considered as pore windows or cavities? Maybe it might be beneficial to use a common terminology in this case.

Furthermore, the authors set their results in context to the work on DUT-49 by Kaskel and co-workers who state:

"...the presence of long-lived metastable states, possibly due to the vast extent of deformation and crossover of pore sizes, compared with the well known, purely microporous 'breathing' (that is, flexible or switchable) systems such as MIL-53. This implies the internal reshuffling of molecules in DUT-49 from rigid MOP pores into contracted octahedral voids."

Interestingly, Kaskel and co-worker do not provide any experimental or computational evidence to support this claim, in contrast to the present study on ZIF-7. The authors may want to elaborate on this a bit more and potentially use this lack of analysis on DUT-49 to motivate their presented

work on ZIF-7.

Furthermore, the authors state that:

"We find that its structural flexibility and stepped CO₂ adsorption isotherms are not a result of the proactive opening of its largest pore upon CO₂ pressure, as described previously in the literature." However, the authors do not cite their own work (J. Mater. Chem. A ,2014,2,620) in which they note that: "A CO₂-induced gate-opening process in D-ZIF-7 is indicated by our experimental results."

After reading this previous work on ZIF-7 - Direct visualisation of carbon dioxide adsorption in gate-opening zeolitic imidazolate framework ZIF-7 (J. Mater. Chem. A ,2014,2,620) I feel that several claims posted in that particular work are in fact supported by the findings in this current study. Why not use this in the introduction part (line 52-54) rather than later in the manuscript to further set the current findings in context to the previous ones.

In line 105 the authors state: "Previously, we studied the CO₂ adsorption geometry in ZIF-7-I at 298 K. according to the reference the experiments were performed at 300 K and the temperature should be corrected."

In general, the following discussion of the results is well written and easy to follow along the very clear figures. The findings are set in context to the previous work and also critically discuss the validity of methods such as Neutron Powder Diffraction for refinement of CO₂ positions via Rietveld refinement.

There are a few questions which I have in regards to the performed experiments:

ZIF-8 is also known to undergo structural transitions under the applied adsorption conditions. Have the authors considered a transition of ZIF-8 to occur under the conditions used in the performed QENS experiments?

the CO₂ self-diffusivity was measured at 225 K while SPXRD and INS analysis were performed at 195 K. Is there an explanation for the difference in temperature that might be worth mentioning in the manuscript?

The presented results nicely illustrate the adsorption mechanism from a unique perspective. Instead of claiming that their work offers opportunities to develop a new family of MOFs with non-uniform dynamic pore characteristics I would rather state that this work allows to analyse adsorption mechanism in other known flexible MOFs since no detailed design principles for other similar materials are provided. If the authors were able to provide a concept of how the observations can be applied in the design of novel flexible MOFs or correlate the findings to phenomena in other known materials it would support the above made statement and generalise the findings of this study.

Although the experiments are performed on a high standard and described in detail, the authors do not provide characterization of the ZIF-7 material used in the study. Providing such experimental analysis of materials properties is in particular important for flexible MOFs since their adsorption behavior is recently found to strongly depend on factors such as crystal size. This has recently been demonstrated by Krause et al. (The effect of crystallite size on pressure amplification in switchable porous solids, Nat Commun. 2018; 9: 1573) on the MOF DUT-49 that the authors state as an example of non-uniform porous structures or ZIF-8 which is structurally related to ZIF-7 (Zhang C, Gee JA, Sholl DS, Lively RP. Crystal-size-dependent structural transitions in nanoporous crystals: adsorption-induced transitions in ZIF-8. J. Phys. Chem. C. 2014;118:20727–20733. ; Tanaka S, et al. Adsorption and diffusion phenomena in crystal size engineered ZIF-8 MOF. J. Phys. Chem. C. 2015;119:28430–28439.). I therefore suggest to provide fundamental data on materials properties by using PXRD and SEM data to estimate the crystal size.

In addition, the authors should check for spelling mistakes (e.g. line 38 though instead of through) and check references for errors e.g. Ref. 11: Y. Du et al., J. Phys. Chem. C, (2017).

In my opinion the introduction part of the manuscript should be significantly restructured to make the paper more accessible also for a broader readership. Additional experimental data on the

materials should be provided to complement the otherwise excellent experimental work. In general, I find the work of great interest but feel that the manuscript does not offer the generality for publication in Nature Communications and thus I suggest a publication in a more specified journal.

Reviewer #2 (Remarks to the Author):

The authors describe the structural change mechanism of ZIF7 upon adsorption of CO₂ by varying temp or pressure using in situ XRD. One uniqueness of this paper is to do a structural characterization at a certain adsorption level to link the two phenomenon together. it is also the first paper that describe the detail structural changes during the second adsorption step in 195K. however, the findings are not that novel given the 195K two step isotherm, the extra large pore ZIF-7 were all disclosed in previous papers before. I will not recommend a publication in nature but perhaps journal like JPC C will be a good fit. a list of questions and technical recommendations are added below.

Page 2 line 58-60, this basic description of ZIF-7 looks a little out of place.

Page 2, line 73-93, this is a lengthy description of some observed phenomenon. It has been well studied that its pore opening is due to the gibbs free energy state which is a combination of its structure and adsorption. As a result, it is always a function of temperature and pressure for a given gas. This is the same reason on page 3, line 100, why CO₂ adsorption happens at a very low pressure at 195K because t such a low pressure, very little (can be as little as 0.001 torr) is enough to trigger the phase transformation from narrow pore to large pore. The fact that when it is absolutely at vacuum it stays as narrow pore also confirms that some CO₂ is needed to cause this initial phase change, regardless whether it is obviously capture in this isotherm reported.

Page 3 105-130, if ZIF-7-II structure is not accurate, how can the authors determine pore B is identical between ZIF-7-I and ZIF-7-II? Also, is the adsorption preferred site B only determined by the fact that the CO₂ diffusivity in ZIF-7 is faster than ZIF-8 hence the pore structure are different? I recall some computational studies indicating that bim linker rotates more and can generate a much bigger opening at a given temperature, could that explain this also?

Reviewer #3 (Remarks to the Author):

The manuscript by P. Zhao et al. reports a non-uniform porous structure ZIF-7 which undergoes a transition induced by CO₂ adsorption. The novelty of this paper is determining the mechanism of pore filling. This is identified via in situ PXRD, QENS and INS experiments. DFT calculations are used to calculate adsorption energies of CO₂ within the two phases.

Overall, I really enjoyed reading the manuscript and find the use of QENS and INS to understand dynamic adsorption behaviour in MOFs very novel.

I feel the paper would be suitable for publication after the following are addressed:

1) The author states in the introduction.

"In general, it is considered that in a flexible MOF all pores capable of gas adsorption behave in a similar way"

I do not find this a generalisation in flexible MOFs. It depends on the topology and size of the pores. There are many examples of multi-sized pore MOFs, which exhibit different adsorption behaviour. E.g Cu-BTC : J. J. Gutiérrez-Sevillano et al., J. Phys. Chem. C, 2013, 117, 21, 11357-11366. The adsorption process of flexible MOFs is driven by different factors such as: adsorption site enthalpy in different pores (which may be driven by size based interactions or types of interactions), flexibility of linker to increase volume, linker rotation or subnetwork displacement. The case where pores would behave the same, will also depend on the symmetry of the transition. In the case of breathing (not the only flexibility seen before in MOFs), these are not symmetry

breaking modes, so all pores will behave in a similar way. In the case described here in this manuscript, I believe you have found a mechanism common to many MOFs. It is one where guest-host interactions are found to heavily influence the “flexibility”. E.g. S. Krause et al. *J. Phys. Chem. C* 2018, 122, 19171–19179, C. Hobday et al., *J. Am. Chem. Soc.* 2017, 140, 1, 382–387 and E. Carrington et al, *Nat. Chem.*, 2017,9, 882–889.

2) I have a question about the isotherm which you present in Figure 1B, and a related question regarding the loading for the QENS and INS experiments.

The author writes: “At 60 kPa, ZIF-7 reaches its full CO₂ adsorption capacity at this temperature (1.3 mmol·g⁻¹).”

The temperature in question is 298 K. However, in the literature, there are many CO₂ isotherms at 298 – 303 K for ZIF-7. The uptake in those is much higher (2.2 mmol g⁻¹ at 100 kPa and 2 mmol g⁻¹ at 60 kPa). E.g. see: Y. Du et al., *J. Am. Chem. Soc.* 2015, 137, 13603–13611; X. Wu et al., *Microporous Mesoporous Mater.*, 2014,190, 189–196; J. van der Bergh et al., *Chem. Eur. J.* 2011, 17, 8832 – 8840; S. Aguado et al., *New J. Chem.*, 2011,35, 546–550.

Do you believe your material is sufficiently activated? Can you be sure there are no residual solvent in the pores which will limit the uptake measured?

Have you also reproduced your isotherm data? Also, a table of pressure/uptake in the SI would make the data more transparent than a small figure.

Leading on from this, for the QENS and INS you state you use a CO₂ loading of 1.3 mmol g⁻¹. How do you determine the loading? Do you assume that it is the same as the maximum adsorbed from the isotherm? It might be more sensible to state a pressure of gas than the uptake. In addition, as the QENS analysis relies on the uptake being measured accurately for measuring the diffusivity (Equation S3), it would be interesting to see how it varies as a function of *c*.

3. I find the DFT section a simplistic view of adsorption energies. Are the authors sure that 10⁻⁴ eV is a sufficient energy convergence. This value is very coarse, especially when looking at such small energy differences between the CO₂ adsorption sites.

The methods section in the SI and manuscript are not detailed enough in order to reproduce the calculation. A text input file provided with the SI would help the reproducibility of the calculations. What is not clear is whether these are optimisations or static energy calculations. Force tolerances are provided, so it is assumed that the crystal structures are optimised. The minimiser should also be included (BFGS, steepest descent... etc.). Do the authors see large movements in the orientation of the CO₂ molecules from the starting crystal structure? Or changes in the orientation of the BIm linkers?

In addition, the equation used to determine the averaged adsorption energy of each CO₂ molecule is too simplistic. $\Delta E = (E - E_f) / N_{CO_2}$. With this equation, you are including the CO₂-CO₂ interaction energy and this will vary depending on the number of CO₂ molecules present and may contribute to why the CO₂ adsorption energy in both A and B pores is 0.146 eV per molecule lower than that in pore A. A more appropriate expression would be: $\Delta E = E - E_f - (E_{CO_2} * N_{CO_2})$. This would remove the contribution of the CO₂ molecules, this quantity should be calculated for increasing numbers of CO₂ molecules in the pores and from that a more accurate adsorption energy would be calculated.

The author should also address that these calculations are carried out at 0 K, and that temperature may have an effect. A more statistically sound way of calculating adsorption energies (with temperature included) is using grand canonical Monte Carlo simulations (S. Jawahery et al. *Nature Commun.* 2017, 8, 13845; J. Hartlieb et al, *J. Am. Chem. Soc.*, 2016, 138 (7), 2292–2301). As it is a statistical mechanics method, a distribution of energies and orientations are calculated. In addition, the method can be used to confirm the uptake of CO₂ in ZIF-7.

Response to Reviewers' Comments

We sincerely appreciate all the valuable comments the reviewers made on our manuscript. Following their constructive advice, we substantially revised the manuscript. Here is our point-to-point response to the concerns raised by the reviewers. We do hope our revision and response meet the high standard set by the reviewers.

Reviewer #1:

In their work "Structural flexibility of a metal-organic framework induced by CO₂ migration in its non-uniform porous structure" Zhao et al. address the structural transition and host-guest interactions in the flexible metal-organic framework (MOF) ZIF-7 by extensive in situ experimental efforts aided by computational analysis. In contrast to most existing "static" studies on flexibility in MOFs, some of them cited herein, this work addresses the adsorption mechanism and flexibility in ZIF-7 by experimental diffusion analysis of the guest, in this case carbon dioxide, which makes it fundamentally interesting for the growing field of stimuli-responsive MOFs. In their findings, the authors refine the adsorption mechanism in ZIF-7 with findings contrary to previous investigations (in part by the authors themselves). In my opinion, the presented work in its current form does not allow the reader to fully understand the fundamental differences to the previously proposed mechanism.

1. In the introduction line 38 the authors state:

"Established theories envisage that when interacting with gas molecules, MOF pores open for adsorption through linker rotation and/or framework distortion." This is only true for a portion of flexible MOFs generally referred to as gate-opening MOFs. Other materials, such as the cited DUT-49 or MIL-53, exhibit contraction of the pores upon interaction with gas molecules. The authors should refine their statement in this regard.

We thank the reviewer for pointing out the incomplete statement we made. We revised the sentence to: "Established theories envisage that when interacting with gas molecules, MOF pores deform for adsorption through linker rotation and/or framework distortion, which leads to the abrupt increase/decrease in gas adsorption capacities."

2. Furthermore, the authors refer to the pressure at which a structural transition occurs as: "adsorption critical pressure". This might be mistaken with the critical pressure of the gas. I suggest changing the applied terminology to transition pressure instead.

We agree with the reviewer that the term "adsorption critical pressure" could be misleading. We have therefore replaced it with "adsorption transition pressure" in the manuscript (highlighted).

3. In regards to the title, the authors state that (line 43):

"In general, it is considered that in a flexible MOF, all pores capable of gas adsorption behave in a similar way. This assumption is used by default because most of the flexible MOFs studied have uniform porous structures."

I feel that in this context the terminology uniform, and later on non-uniform porous structure, might not be an adequate description and potentially misleading for readers. To avoid confusion, the authors should provide detailed example of what they understand to be

uniform and non-uniform pore structures instead of just citing three recent reviews about flexible MOFs in general (reference 7-9) to support their definition.

The authors describe DUT-49, a MOF that exhibits a pore structure consisting of micro and mesopores, as non-uniform porous structure and set the investigated ZIF-7 in context to DUT-49. Thus, a MOF with channelled pore structure such as MIL-53 exhibits a uniform pore structure?

Because this terminology is used in the title, I feel a more detailed definition would support the readability of the manuscript.

From the reviewer's feedback, we realized that we didn't manage to clearly explain the meaning of "uniform" and "non-uniform" porous structure. Thus we have given a more detailed definition of these terms in the revised manuscript:

"This assumption is used by default because most of the flexible MOFs studied in detail have uniform porous structures with only one type of pores that have the same size, shape, and environment (7-9). For example, MIL-53 (10), one of the most studied flexible MOFs, shows only one type of quadrilateral prism pores of 8.5 Å diameter (11). In recent years, MOFs with non-uniform porous structures, i.e. with multiple types of pores which vary in size, shape, and environment, have been shown to also have structural flexibility and unique gas adsorption features (12)."

4. To further define a non-uniform pore structure, the authors can present pores size distribution analysis of ZIF-8. In fact, I believe it would strengthen the proposed mechanism if an evolution of the pore size distribution in ZIF-7-I and ZIF-7-II would be provided and set in context to the investigated adsorption sites.

Following the reviewer's suggestion, we performed pores size distribution analysis of ZIF-8, ZIF-7-I, and ZIF-7-II in CrystalMaker 10.3 using their crystal structures determined by diffraction methods (ZIF-8 and ZIF-7-I: K. S. Park et al., Proc. Natl. Acad. Sci. USA 2006, 103, 10186-10191; ZIF-7-II: P. Zhao et al., Chem. Mater. 2014, 26, 1767-1769). The probe radius was 1.55 Å, the same as the van der Waals radius of a N₂ molecule. Results are presented below as well as in Fig. S1 and Table S1.

From the analysis results, it is clear that there is only one type of pores in ZIF-8 (7.1 Å), which corresponds to the sodalite cage of ZIF-8. This indicates that ZIF-8 has a uniform porous structure. In ZIF-7-I, four types of pores with distinct radius were found: pore A (4.0 Å), sodalite cage (SOD, 3.8 Å), pore B (3.2 Å), and four-member-ring pore (4M, 2.4 Å). The pore distribution in ZIF-7-II is very different from that in ZIF-7-I. During ZIF-7-II to ZIF-7-I phase transition, the radius of pore B decrease from 3.5 – 4.7 Å to 3.2 Å, whereas the radii of pore A and sodalite cage increase from 2.3 – 2.5 Å and 2.8 – 3.1 Å to 4.0 Å and 3.8 Å, respectively. The radius of four-member-ring pore remains unchanged (2.5 – 2.9 Å to 2.4 Å).

As ZIF-7-II to ZIF-7-I phase transition is induced by CO₂ adsorption, it can be said that with the filling of CO₂, pore B decreases in size whereas pore A increases in size. This strengthens our proposed mechanism on the sequential CO₂ filling of pore B (first) and pore A (last). The filling of pore A with CO₂ requires pore deformation which is induced by the interaction between filling CO₂ and pore B.

We have added the above discussion under Fig. S1.

In this context, the authors state: "There are four types of pores in ZIF-7, including two types of six-member-ring pores, one type of four-member-ring pores, and the Sodalite cage." In previous literature on ZIF-7 these pores are usually considered as pore windows or cavities? Maybe it might be beneficial to use a common terminology in this case.

Regarding the terminology, the reviewer is right that ZIF-7 pores defined in this manuscript were usually described as pore windows or cavities in early literature. This is because the

sodalite cage of ZIF-7 was first described as the main pore by K. S. Park et al., Proc. Natl. Acad. Sci. USA 2006, 103, 10186-10191. However, we think that this description is out of date as there are many recent papers discussing the different adsorption pores/cavities in ZIF-7. In addition, this previous description makes it difficult for us to present our concept of non-uniform porous structure. Thus we keep the present terminology in the revised manuscript.

5. Furthermore, the authors set their results in context to the work on DUT-49 by Kaskel and co-workers who state:

"...the presence of long-lived metastable states, possibly due to the vast extent of deformation and crossover of pore sizes, compared with the well-known, purely microporous 'breathing' (that is, flexible or switchable) systems such as MIL-53. This implies the internal reshuffling of molecules in DUT-49 from rigid MOP pores into contracted octahedral voids."

Interestingly, Kaskel and co-worker do not provide any experimental or computational evidence to support this claim, in contrast to the present study on ZIF-7. The authors may want to elaborate on this a bit more and potentially use this lack of analysis on DUT-49 to motivate their presented work on ZIF-7.

We are pleased with the clear recognition from this reviewer of our contribution and thank him/her for the suggestion to differentiate our work from those in the literature. Following the suggestion, we have included the discussion on the statement from Kaskel et al. in the introduction section of the revised manuscript:

"Noticeably, although this work indicates the internal shuffling of gas molecules between different pores of DUT-49, this claim was unfortunately not supported by any dynamic study. It motivated us to perform a detailed dynamic analysis of guest molecules inside a flexible non-uniform porous MOF. We believe that this solves a fundamentally important issue in the understanding of structural transitions and guest-host interactions in flexible MOFs."

6. Furthermore, the authors state that:

"We find that its structural flexibility and stepped CO₂ adsorption isotherms are not a result of the proactive opening of its largest pore upon CO₂ pressure, as described previously in the literature."

However, the authors do not cite their own work (J. Mater. Chem. A ,2014,2,620) in which they note that: "A CO₂-induced gate-opening process in D-ZIF-7 is indicated by our experimental results."

After reading this previous work on ZIF-7 - Direct visualisation of carbon dioxide adsorption in gate-opening zeolitic imidazolate framework ZIF-7 (J. Mater. Chem. A ,2014,2,620) I feel that several claims posted in that particular work are in fact supported by the findings in this current study. Why not use this in the introduction part (line 52-54) rather than later in the manuscript to further set the current findings in context to the previous ones.

In our previous work (J. Mater. Chem. A, 2014, 2, 620), we discussed the opening of ZIF-7 pores after the phase transition (in ZIF-7-I phase). This phenomenon is due to pore overfilling with CO₂ molecules. We felt the discussion of this opening behavior was not quite relevant to what we would like to discuss in this manuscript: the gate-opening phenomenon

related to the phase transition. Thus we did not cite our own work. However, as the reviewer suggested, we should include our own work to further set the current findings in context to the previous ones. In our previous work (J. Mater. Chem. A, 2014, 2, 620), we did comment on the importance of CO₂ migration inside the structure upon pressure. However, the lack of evidence at that time prohibited us from giving a clear description of the migration route until now. We use this as one of the motivations for the present work:

“Type A six-member-ring pore (pore A) has the largest void space in ZIF-7 and has long been considered the one to proactively “open” for gas adsorption (15-17). However, we found that the structural flexibility and stepped CO₂ adsorption isotherms of ZIF-7 is in fact related to CO₂ migration between its different types of pores, something that we implied in our previous work (18). Although formerly we were prohibited from giving a clear description of the migration route due to the lack of evidence, in the present work we are able to include extensive in situ experimental and computational analysis revealing the CO₂ migration pathways and its relationship with the adsorption-stimulated structural transitions in ZIF-7.”

7. In line 105 the authors state: "Previously, we studied the CO₂ adsorption geometry in ZIF-7-I at 298 K. according to the reference the experiments were performed at 300 K and the temperature should be corrected."

We thank the reviewer for pointing out the mistake we made in the original manuscript. We have corrected the temperature accordingly (highlighted).

In general, the following discussion of the results is well written and easy to follow along the very clear figures. The findings are set in context to the previous work and also critically discuss the validity of methods such as Neutron Powder Diffraction for refinement of CO₂ positions via Rietveld refinement.

There are a few questions which I have in regards to the performed experiments:

8. ZIF-8 is also known to undergo structural transitions under the applied adsorption conditions. Have the authors considered a transition of ZIF-8 to occur under the conditions used in the performed QENS experiments?

We understand the concern raised by the reviewer. The diffusivity of ZIF-8 cited in this manuscript was obtained from the following references:

24. C. Chmelik, J. van Baten, R. Krishna, J. Membr. Sci. 397-398, 87-91 (2012).

25. E. Pantatosaki et al., J. Phys. Chem. C 116, 201-207 (2012).

26. H. Bux, C. Chmelik, J. M. van Baten, R. Krishna, J. Caro, Adv. Mater. 22, 4741-4743 (2010).

CO ₂ loading (molecules/unit cell)	T (K)	D _t (10 ⁻⁹ m ² ·s ⁻¹)	Method	Reference
0 – 10	298	0.1 – 0.7	Infrared microscopy	24, 26
0 – 10	303	0.3 – 1.1	Molecular dynamics simulation	25

In these references, a phase transition in ZIF-8 was not considered due to the low CO₂ loading range. According to T. Chokbunpiam et al., Chem. Phys. Lett. 2016, 648, 178-181, at 298 K, CO₂ adsorption induces the phase transition in ZIF-8 at a CO₂ loading higher than 20 molecules/unit cell, which is beyond the range in ref 24-26. T. Chokbunpiam et al. also

mentioned that after the phase transition in ZIF-8, dramatic decrease in CO₂ diffusivity was found by their molecular dynamics simulation. This may be due to the dense packing of CO₂ molecules inside the pores. In addition, at the highest CO₂ loading (30 molecules/unit cell), the window size of ZIF-8 CO₂ hosting pore is 4.1 Å. The geometry of its window is unchanged and still resembles that of pore A in the ZIF-7-I phase.

Thus it is believed that the phase transition in ZIF-8 does not take place under the cited conditions and will not affect the conclusion in our manuscript:

“The measured CO₂ transport diffusivity in ZIF-7-I ($6.3(7) \times 10^{-9} \text{ m}^2 \cdot \text{s}^{-1}$, Fig. 2B, see SI for detailed analysis) was compared with those experimentally obtained in ZIF-8 at the same temperature ($0.4(3) \times 10^{-9} \text{ m}^2 \cdot \text{s}^{-1}$) (23-26). We did this comparison because ZIF-8 is an analog of ZIF-7 but has only one type of CO₂ hosting pore (see pore size distribution analysis in Fig. S1 and Table S1) connected by six-member-ring windows with similar size and geometry (diameter 3.4 – 4.1 Å) to those of pore A in ZIF-7-I (Fig. S3) (14, 23). In contrast, pore B in ZIF-7-I and ZIF-7-II has larger windows of 4.6 – 5.3 Å diameter. The one-order-of-magnitude larger CO₂ transport diffusivity in ZIF-7-I compared to ZIF-8 indicates that, in accessing ZIF-7-I, CO₂ prefers pore B to pore A.”

As the diffusivity presented was based on the experimental data in ref 24 and 26, we changed the first sentence to: “...was compared with those experimentally obtained in ZIF-8 at the same temperature...” We also cited T. Chokbunpiam et al., Chem. Phys. Lett. 2016, 648, 178-181.

9. The CO₂ self-diffusivity was measured at 225 K while SPXRD and INS analysis were performed at 195 K. Is there an explanation for the difference in temperature that might be worth mentioning in the manuscript?

We used 225 K for QENS so that CO₂ diffusivities are above the detection limit. 195 K is the boiling point of CO₂, at this temperature CO₂ diffusion is too slow to be measured accurately. For SPXRD, 195 K was used so that we can monitor the structural evolution of ZIF-7 and compare the results with the adsorption isotherm measured at the same temperature. INS data was collected at below 10 K to minimize the thermal motion of CO₂ molecules and the host framework which influences the accuracy of the measurement. Namely, at higher temperatures, the increase in Debye-Waller factor causes broadening of the spectral peaks, i.e. a decrease in peak intensity or spectral resolution. We have included the above explanation in the revised manuscript and SI.

10. The presented results nicely illustrate the adsorption mechanism from a unique perspective. Instead of claiming that their work offers opportunities to develop a new family of MOFs with non-uniform dynamic pore characteristics I would rather state that this work allows analysing adsorption mechanism in other known flexible MOFs since no detailed design principles for other similar materials are provided. If the authors were able to provide a concept of how the observations can be applied in the design of novel flexible MOFs or correlate the findings to phenomena in other known materials it would support the above made statement and generalise the findings of this study.

We agree with the reviewer on generalizing our adsorption mechanism, hence leading to design principles for optimizing other flexible MOFs with non-uniform porous structures. In light of the reviewer’s comment, we revised the final sentence of the conclusion part:

“Overall, the relationship between guest and host dynamics discussed here provides new insights into MOF flexibility. This work also offers new methods for analyzing adsorption mechanisms in other known flexible MOFs, particularly those with non-uniform porous structures, by evaluating the adsorption strength in each type of pores and the extent of guest migration with respect to the phase change. This may lead to the selective filling of a specific type of pores without triggering the associated phase transformation, which could result in the rational design of optimized stimuli-responsive behaviors.”

11. Although the experiments are performed on a high standard and described in detail, the authors do not provide characterization of the ZIF-7 material used in the study. Providing such experimental analysis of materials properties is in particular important for flexible MOFs since their adsorption behavior is recently found to strongly depend on factors such as crystal size. This has recently been demonstrated by Krause et al. (The effect of crystallite size on pressure amplification in switchable porous solids, *Nat Commun.* 2018; 9: 1573) on the MOF DUT-49 that the authors state as an example of non-uniform porous structures or ZIF-8 which is structurally relate to ZIF-7 (Zhang C, Gee JA, Sholl DS, Lively RP. Crystal-size-dependent structural transitions in nanoporous crystals: adsorption-induced transitions in ZIF-8. *J. Phys. Chem. C.* 2014;118:20727–20733. ; Tanaka S, et al. Adsorption and diffusion phenomena in crystal size engineered ZIF-8 MOF. *J. Phys. Chem. C.* 2015;119:28430–28439.). I therefore suggest providing fundamental data on materials properties by using PXRD and SEM data to estimate the crystal size.

We agree with the reviewer that some basic characterization of ZIF-7 material is missing in the original manuscript. The reviewer mentioned that since crystal size heavily influences the adsorption behaviors of flexible MOFs, related material characterization should be presented. Thus, we added the PXRD data and SEM images of an as-synthesized ZIF-7 sample and an activated ZIF-7 sample in the SI (Figs. S9–S11) to give the readers a good idea of the crystal size of our samples used for advanced characterization. According to SEM images, the average crystal size of the as-synthesized ZIF-7 sample is estimated to be around 17 μm . The average crystal size of the activated ZIF-7 sample is estimated to be around 2 μm . The activated ZIF-7 sample is in ZIF-7-II phase and is expected to have a much smaller crystal size than the as-synthesized ZIF-7 sample which is in ZIF-7-I phase. This is because the PXRD peaks of ZIF-7-II are much broader than those of ZIF-7-I (Fig. S9). Crystal size estimation based on PXRD data using the Scherrer equation was not performed because this method is only suitable for crystals smaller than 0.1 μm . We read the references mentioned by the reviewer thoroughly. These references indicate that for DUT-49 and ZIF-8 with nanometer-scaled crystal sizes, their adsorption-induced flexibility is suppressed or the transition pressure is significantly higher than that for those with micrometer-scaled crystal sizes. Our activated ZIF-7 sample contains micrometer-scaled crystals. This allows it well exhibiting adsorption-induced flexibility under our experimental conditions. The crystal sizes of all activated samples analyzed by various techniques in this manuscript are the same, so the crystal size effect will not be discussed in the main text. We have added this discussion under Fig. S11.

12. In addition, the authors should check for spelling mistakes (e.g. line 38 though instead of through) and check references for errors e.g. Ref. 11: Y. Du et al., *J. Phys. Chem. C*, (2017).

We thank the reviewer for kindly helping us check mistakes and errors. We have corrected the mistakes mentioned by the reviewer: Ref. 21: Y. Du et al., *J. Phys. Chem. C*, 121,

28090-28095 (2017). We also used software such as Grammarly to check for spelling mistakes, hopefully, there is no mistake anymore.

In my opinion the introduction part of the manuscript should be significantly restructured to make the paper more accessible also for a broader readership. Additional experimental data on the materials should be provided to complement the otherwise excellent experimental work. In general, I find the work of great interest but feel that the manuscript does not offer the generality for publication in Nature Communications and thus I suggest a publication in a more specified journal.

We thank the reviewer for his/her critical but very useful comments on this manuscript. As seen in the revised manuscript, we have restructured the introduction part. The novelty of this work is stressed on the demonstration of a new gas adsorption mechanism under dynamic conditions. The mechanism involves the sequential filling of ZIF-7's multiple types of pores and the internal shuffling of CO₂ molecules between these pores. It is significantly different from those widely discussed in the literature for MOF structures that contain only one type of guest-hosting pores. We hope this new adsorption mechanism will guide the synthesis of flexible MOFs for the optimization of their stimuli-responsive behaviors. We have also provided additional experimental and computational data to support our conclusions.

Reviewer #2:

The authors describe the structural change mechanism of ZIF7 upon adsorption of CO₂ by varying temp or pressure using in situ XRD. One uniqueness of this paper is to do a structural characterization at a certain adsorption level to link the two phenomena together. It is also the first paper that describes the detail structural changes during the second adsorption step in 195K. However, the findings are not that novel given the 195K two step isotherm, the extra large pore ZIF-7 were all disclosed in previous papers before. I will not recommend a publication in nature but perhaps journal like JPC C will be a good fit. A list of questions and technical recommendations are added below.

We thank the comment from the reviewer. However, we feel that there must be some misunderstanding from this reviewer on our manuscript, presumably due to some unclearness in our original manuscript. We have now substantially rewritten our manuscript to clarify a few points and to stress a bit more on the novelty of this work.

First, we did not claim that we made the original observation of the two-step isotherm at 195K nor the extra-large pore phase (the relevant papers were cited). Instead, we studied for the first time the adsorption properties of ZIF-7 experimentally and theoretically under dynamic conditions.

The novelty of this work is the demonstration of a new gas adsorption mechanism under dynamic conditions. The mechanism involves the sequential filling of ZIF-7's multiple types of pores and the internal shuffling of CO₂ molecules between these pores. It is significantly different from those widely discussed in the literature for MOF structures that contain only one type of guest-hosting pores. This new adsorption mechanism will guide the synthesis of flexible MOFs for the optimization of their stimuli-responsive behaviors. As a result, with

substantial modifications, we hope that this reviewer can appreciate the merits of this work, which can be recognized by the other two reviewers.

1. Page 2 line 58-60, this basic description of ZIF-7 looks a little out of place.

We thank the reviewer for pointing out our description of ZIF-7 was not adequate. We made improvement in the revised manuscript:

“Here, we decide to study zeolitic imidazolate framework 7 (ZIF-7) which has a non-uniform porous structure showing structural flexibility and stepped adsorption isotherms during the uptake of guest molecules such as CO₂ (13). It is an important member of the MOF subgroup, ZIFs, some of which are noted for their high hydrothermal stability and structural similarity to aluminosilicate zeolites (14). ZIF-7 (Zn(bIm)₂, bIm: benzimidazolate) is composed of zinc atoms connected by bIm linkers in tetrahedral coordination. As indicated by the pore size distribution analysis (Fig. S1 and Table S1), ZIF-7 has four types of pores, including two types of six-member-ring pores, one type of four-member-ring pores, and the sodalite cage (Fig. 1A).”

2. Page 2, line 73-93, this is a lengthy description of some observed phenomenon. It has been well studied that its pore opening is due to the Gibbs free energy state which is a combination of its structure and adsorption. As a result, it is always a function of temperature and pressure for a given gas.

Lines 73-93 on page 2 are used to describe the CO₂ adsorption isotherms of ZIF-7 at 195 and 298 K. While it is true that the pore opening of ZIF-7 (a phase change from ZIF-7-II to ZIF-7-I) at 298 K is simply related to the Gibbs free energy state of the system which is dependent on gas pressure, as seen from the 195 K data, a new stepped adsorption at 0-10 kPa can be obtained but without phase change (maintained in ZIF-7-II, as indicated by our SXRD data). Such an observation cannot be rationalized using only one type of adsorption site. Discussions in our manuscript clearly suggest the sequential filling of CO₂ in pore B and then in pore A due to high kinetic barriers unless higher pressures are used (then associated with the phase change to ZIF-7-I). Thus, it is inadequate to only using the thermodynamics to describe the structural flexibility and adsorption mechanism of ZIF-7 at low temperatures without involving two adsorption pores. In addition, one is unable to directly correlate the ZIF-7-II to ZIF-7-I phase transition to CO₂ migration from one pore to another. Thus, in this paper, we performed detailed structural and diffusion studies for CO₂ migration route between two different adsorption pores and its relationship with structural flexibility and adsorption behavior of ZIF-7.

This is the same reason on page 3, line 100, why CO₂ adsorption happens at a very low pressure at 195K because at such a low pressure, very little (can be as little as 0.001torr) is enough to trigger the phase transformation from narrow pore to large pore. The fact that when it is absolutely at vacuum it stays as narrow pore also confirms that some CO₂ is needed to cause this initial phase change, regardless whether it is obviously capture in this isotherm reported.

We thank this reviewer for this comment but feel he/she might take a simple view: at low CO₂ partial pressures a narrow-pore phase results but at high CO₂ partial pressures a large pore is established. However, as seen from the adsorption isotherm at 195K, there is an extra step change at 0-10 kPa compared with 298K. This clearly doesn't indicate a

progressive pore swelling at increasing CO₂ loading. Besides, our SXR D data show that at 195 K, the phase transition did not take place in this pressure range (maintained in ZIF-7-II). Discussions in our manuscript indicate that at such low temperature CO₂ can be adsorbed in pore B of ZIF-7-II without being rapidly transferred to pore A. The absence of this observation at 298 K can be rationalized by the fact that one cannot fill pore B of ZIF-7-II by CO₂ molecules at higher temperatures as CO₂ molecules cannot be stabilized by their weak hydrogen bonding to the framework.

We also performed additional DFT and grand canonical Monte Carlo (GCMC) molecular simulations, in order to demonstrate that substantial CO₂ uptake/pressure (far more than 0.001 torr) is needed for the phase transition to take place at 195 K.

The above figure shows the heat of adsorption (Q_{st}) as a function of CO₂ uptake (obtained from GCMC simulations). Triangles: ZIF-7-I, squares: ZIF-7-II. Orange: 195 K, purple: 298 K. Red dotted lines are included as eye-guides.

DFT calculations estimate that the energy difference between ZIF-7-I and ZIF-7-II framework (ΔE_f) is around 29.9 kJ·mol⁻¹ (one unit cell). Upon adsorption, this energy difference is expected to be overcome by the affinity of CO₂ in pore B of ZIF-7-II. The difference in the heat of adsorption of CO₂ (ΔQ_{st}) in the transition region can be given by (S. Kaskel et al., Nature 2016, 532, 348)

$$\Delta Q_{st} = N'_{CO_2} \times (Q_{st-I} - Q_{st-II})$$

where N'_{CO_2} is the number of CO₂ molecules adsorbed in one ZIF-7 unit cell, Q_{st-I} and Q_{st-II} are the Q_{st} obtained for ZIF-7-I and ZIF-7-II, respectively. At 298 K, ΔQ_{st} is 30.1 kJ·mol⁻¹ at the uptake of 1.3 mmol·g⁻¹ ($N_{CO_2} = 7$) where the ZIF-7-II to ZIF-7-I phase transition completes. This value is comparable to ΔE_f . At 195 K, maximum ΔQ_{st} is 34.5 kJ·mol⁻¹ at the uptake of 2 mmol·g⁻¹ ($N_{CO_2} = 11$), also comparable to ΔE_f . These results underpin that substantial initial adsorption is needed to compensate for the energy penalty of the structural transition.

We have included this discussion in the main text.

3. Page 3 105-130, if ZIF-7-II structure is not accurate, how can the authors determine pore B is identical between ZIF-7-I and ZIF-7-II?

We imagine there might be a misunderstanding about the structure determination. In the manuscript, we commented that it is difficult to determine the CO₂ locations in ZIF-7-II at 195 K using diffraction methods. This is because ZIF-7-II has low symmetry (P-1) and a large unit cell ($V = 7917 \text{ \AA}^3$), Rietveld refinement based on diffraction data is not able to define a reliable ZIF-7-II (CO₂) crystal structure.

However, the parent structure (ZIF-7-II without CO₂) has been established with confidence in our previous peer-reviewed paper (Chem. Mater. 2014, 26, 1767-1769). Using this parent structure as a model, one can establish the fact that pore B is identical between ZIF-7-I and ZIF-7-II

Also, is the adsorption preferred site B only determined by the fact that the CO₂ diffusivity in ZIF-7 is faster than ZIF-8 hence the pore structure are different? I recall some computational studies indicating that blm linker rotates more and can generate a much bigger opening at a given temperature, could that explain this also?

The adsorption preferred site B is determined by high-resolution neutron powder diffraction in our previous work (J. Mater. Chem. A ,2014, 2, 620-623). This conclusion is confirmed by works from others, e.g. C. Cuadrado-Collados et al., J. Mater. Chem. A 2017, 5, 20938-20946; Y. Du et al., J. Phys. Chem. C 2017, 121, 28090-28095; Y. Du et al., J. Am. Chem. Soc. 2015, 137, 13603-13611. The existence of a ZIF-7 phase having pores with a much bigger opening is not evident in any of the diffraction data presented in this work. Thus, we did not consider the extra-large pore phase.

Reviewer #3:

The manuscript by P. Zhao et al. reports a non-uniform porous structure ZIF-7 which undergoes a transition induced by CO₂ adsorption. The novelty of this paper is determining the mechanism of pore filling. This is identified via in situ PXRD, QENS and INS experiments. DFT calculations are used to calculate adsorption energies of CO₂ within the two phases.

Overall, I really enjoyed reading the manuscript and find the use of QENS and INS to understand dynamic adsorption behaviour in MOFs very novel.

I feel the paper would be suitable for publication after the following are addressed:

1. The author states in the introduction:

“In general, it is considered that in a flexible MOF all pores capable of gas adsorption behave in a similar way”

I do not find this a generalisation in flexible MOFs. It depends on the topology and size of the pores. There are many examples of multi-sized pore MOFs, which exhibit different adsorption behaviour. E.g Cu-BTC : J. J. Gutiérrez-Sevillano et al., J. Phys. Chem. C, 2013, 117, 21, 11357-11366. The adsorption process of flexible MOFs is driven by different factors

such as: adsorption site enthalpy in different pores (which may be driven by size based interactions or types of interactions), flexibility of linker to increase volume, linker rotation or subnetwork displacement. The case where pores would behave the same will also depend on the symmetry of the transition. In the case of breathing (not the only flexibility seen before in MOFs), these are not symmetry breaking modes, so all pores will behave in a similar way. In the case described here in this manuscript, I believe you have found a mechanism common to many MOFs. It is one where guest-host interactions are found to heavily influence the “flexibility”. E.g. S. Krause et al. *J. Phys. Chem. C* 2018, 122, 19171–19179, C. Hobday et al., *J. Am. Chem. Soc.* 2017, 140, 1, 382–387 and E. Carrington et al, *Nat. Chem.*, 2017,9, 882–889.

From the reviewer’s feedback, we realized that the original sentence we wrote is misleading. It seems that the claim the reviewer got from our statement was that the adsorption behavior of all pores in a flexible MOF is the same. This is, as the reviewer pointed out, not true. However, what we wanted to express here is that, in most studies about a flexible MOF that undergoes a phase transition during gas adsorption, the structural change discovered for all the pores are similar. This is because in these studies, the MOF studied has a uniform porous structure. Thus we revised our statement to:

“In most cases, it is considered that in a flexible MOF, all pores capable of gas adsorption undergo a similar structural change. This assumption is used by default because most of the flexible MOFs studied in detail have uniform porous structures with only one type of pores that have the same pore size, shape, and environment (7-9).”

2. I have a question about the isotherm which you present in Figure 1B, and a related question regarding the loading for the QENS and INS experiments.

The author writes: “At 60 kPa, ZIF-7 reaches its full CO₂ adsorption capacity at this temperature (1.3 mmol·g⁻¹).”

The temperature in question is 298 K. However, in the literature, there are many CO₂ isotherms at 298 – 303 K for ZIF-7. The uptake in those is much higher (2.2 mmol g⁻¹ at 100 kPa and 2 mmol g⁻¹ at 60 kPa). E.g. see: Y. Du et al., *J. Am. Chem. Soc.* 2015, 137, 13603–13611; X. Wu et al., *Microporous Mesoporous Mater.*, 2014,190, 189–196; J. van der Bergh et al., *Chem. Eur. J.* 2011, 17, 8832 – 8840; S. Aguado et al., *New J. Chem.*, 2011,35, 546–550.

Do you believe your material is sufficiently activated? Can you be sure there are no residual solvent in the pores which will limit the uptake measured? Have you also reproduced your isotherm data? Also, a table of pressure/uptake in the SI would make the data more transparent than a small figure.

Before activating the as-synthesized ZIF-7 sample, TG analysis was conducted to monitor the solvent loss process upon heating. By doing so, we decided the activation temperature. The sample was heated from 308 to 573 K at 5 K·min⁻¹ under N₂. Around 350 K, the onset of solvent loss is seen in the TG trace below.

Thus, the as-synthesized sample was activated at 400 K in air for 48 h. Strongly-bonded DMF or methanol (solvent) molecules should have been removed. A combined TG–FTIR analysis was conducted to monitor the guest loss process in the activated sample. FTIR data show no evidence of any residual DMF or methanol molecule in the activated sample. Before adsorption measurement, the sample was vacuumed in situ ($\sim 10^{-4}$ mbar) at room temperature for 3 h to get rid of any remaining DMF, methanol or water molecules.

We compared our activation method with those in the references mentioned by the reviewer.

1) Y. Du et al., *J. Am. Chem. Soc.* 2015, 137, 13603-13611: The solid product washed thoroughly with acetonitrile (~ 90 mL \times 3) and stored in acetonitrile. Prior to each adsorption experiment, all activated ZIF-7 samples reported were subjected to in-situ outgassing for 6 hours at room temperature under vacuum of the order of 1×10^{-4} Torr.

2) X. Wu et al., *Microporous Mesoporous Mater.*, 2014, 190, 189-196: The guest molecules incorporated in the crystals were removed under a dynamic vacuum at 150 °C (423 K) for 12 h. For the adsorption measurement, the initial degassing process was carried out at 150 °C (423 K) for 12 h under a 0.0001 mmHg vacuum pressure.

3) J. van der Bergh et al., *Chem. Eur. J.* 2011, 17, 8832-8840: Prior to the adsorption measurements the adsorbent particles were outgassed in situ under vacuum at 448 K for 16 h to remove any adsorbed impurities.

4) S. Aguado et al., *New J. Chem.*, 2011, 35, 546-550: The samples were outgassed under vacuum ($\sim 10^{-4}$ mbar) at 473 K for 12 h before beginning the measurements.

It is clear that our activation conditions are similar to those reported in the literature. It is known that ZIF-7 should be in ZIF-7-II phase when no or little residual guest molecule is in the pores (S. Aguado et al., *New J. Chem.*, 2011, 35, 546-550). We checked the PXRD data of our activated sample (Fig. 2A and Fig. S9B) and confirmed that the sample was in ZIF-7-II phase. This indicates that our activated sample should not contain much residual solvent that affects its adsorption capacity.

We were at first, as the reviewer did, surprised by the fact that the adsorption capacity of our sample was not on par with the reported values. Thus, we performed the same adsorption measurement on samples activated at higher temperatures and using the gas loading systems at the beamlines where we did QENS and INS experiments. The results were, however, consistent with the reported data in this manuscript.

Very recent studies on the CO₂ adsorption over ZIF-7 show similar capacity as we measured (1.7 mmol g⁻¹ at 100 kPa):

- 1) C. Cuadrado-Collados et al., J. Mater. Chem. A, 2017, 5, 20938-20946: 1.8 mmol g⁻¹ at 100 kPa after the abrupt increase in CO₂ adsorption
- 2) A. Arami-Niya et al., J. Mater. Chem. A, 2017, 5, 21389-21399: 1.6 mmol g⁻¹ at 100 kPa after the abrupt increase in CO₂ adsorption

In order to avoid confusion, we corrected the statement we made in the original manuscript to: “At 60 kPa, ZIF-7 reaches a much higher CO₂ adsorption capacity (1.3 mmol·g⁻¹).” Following the reviewer’s suggestion, we added a table of pressure/uptake (Table S2) to make the data more transparent.

Leading on from this, for the QENS and INS you state you use a CO₂ loading of 1.3 mmol g⁻¹. How do you determine the loading? Do you assume that it is the same as the maximum adsorbed from the isotherm? It might be more sensible to state a pressure of gas than the uptake.

The CO₂ loadings in the QENS and INS experiments were measured at 298 K using the method employed in Z. Lu et al., Nat. Commun. 2017, 8, 14212; S. Yang et al., Nat. Chem. 2014, 7, 121; S. Yang et al., Nat. Chem. 2012, 4, 887-894.

The volume of the system without the sample cell (i.e. to the valve on top of the cell) (V_1) was measured using N_2 . A certain amount of N_2 (n_0) was loaded into the vacuumed system at room temperature so that a pressure of p_0 was reached (shown on the barometer). According to the ideal gas law, we can obtain the value of V_1 :

$$p_0V_1 = n_0RT$$

The volume of the system including the sample cell (V_2) was also measured using the same method. ZIF-7 has almost no N_2 adsorption capacity at 298 K (S. Aguado et al., New J. Chem., 2011,35, 546-550), so V_2 is the actual volume of the system. After the sample was evacuated to 10^{-5} Pa at room temperature for 16 h to remove any remaining trace guest molecules, the valve on top of the sample cell was closed. CO_2 was then loaded into the system to reach a pressure of p_1 . The amount of CO_2 in the system (n_1) can be calculated:

$$p_1V_1 = n_1RT$$

Then the valve on top of the sample cell was open, the CO_2 pressure will stabilize at p_2 . The amount of CO_2 left in the system (n_2) can also be calculated:

$$p_2V_2 = n_2RT$$

The amount of CO_2 adsorbed (n) by ZIF-7 is:

$$n = n_1 - n_2$$

As we knew the amount of sample used in each experiment (ca. 4.6 g), we could precisely control the CO_2 loading to be 1.3 mmol g^{-1} .

Because of the method we used for gas loading, it is most sensible to use number of moles to present the uptake and to compare with that in the adsorption isotherm measured by a gravimetric analyzer.

The above description has been added to SI to give the readers a better understanding of the experimental procedure.

In addition, as the QENS analysis relies on the uptake being measured accurately for measuring the diffusivity (Equation S3), it would be interesting to see how it varies as a function of c .

As mentioned earlier, using the gas loading system presented, we were able to precisely control the CO_2 loading and directly compare the value with that in the adsorption isotherm measured by a gravimetric analyzer. Thus the thermodynamic correction factor ($\partial \ln p / \partial \ln c$) in Equation S7 can be calculated.

When performing the QENS experiment, we intended to vary c to see how the diffusivity would change. However, as we referred to the adsorption isotherms reported by S. Aguado et al., New J. Chem., 2011,35, 546-550 (which indicate ZIF-7 should have a larger adsorption capacity than what we measured), we saturated our sample in the first trial. Due to the time limit at the beamline, we were not able to start the experiment again. Thus, unfortunately, we are not able to provide the relationship between diffusivity and c . We do appreciate that the reviewer pointed out this topic is interesting and we would like to carry out a related study in the future.

3. I find the DFT section a simplistic view of adsorption energies. Are the authors sure that 10⁻⁴ eV is a sufficient energy convergence. This value is very coarse, especially when looking at such small energy differences between the CO₂ adsorption sites.

Thanks for the reviewer's comments. We repeated our DFT calculations using a much stringent criterion which is 10⁻⁶ eV for energy convergence in the self-consistent electronic relaxation. The shape of the unit cell and atomic positions were allowed to fully relaxed during the optimization with the force convergence set to be 0.01 eV·Å⁻¹. We have added this description in the Method section of the revised manuscript.

The methods section in the SI and manuscript are not detailed enough in order to reproduce the calculation. A text input file provided with the SI would help the reproducibility of the calculations. What is not clear is whether these are optimisations or static energy calculations. Force tolerances are provided, so it is assumed that the crystal structures are optimised. The minimiser should also be included (BFGS, steepest descent... etc.). Do the authors see large movements in the orientation of the CO₂ molecules from the starting crystal structure? Or changes in the orientation of the Blm linkers?

We included a typical input file "INCAR" of our DFT calculations in the supporting information. Details of the calculation can be found in this file, for examples we used BFGS as the minimizer. Comparing the starting and optimized crystal structure, we hardly found any major difference. It was only found that CO₂ molecules are localized near the blm linkers after energy relaxation. This is in consistency with the experimental results. We have added this discussion in the revised manuscript.

In addition, the equation used to determine the averaged adsorption energy of each CO₂ molecule is too simplistic. $\Delta E = (E - E_f) / N_{CO_2}$. With this equation, you are including the CO₂-CO₂ interaction energy and this will vary depending on the number of CO₂ molecules present and may contribute to why the CO₂ adsorption energy in both A and B pores is 0.146 eV per molecule lower than that in pore A. A more appropriate expression would be: $\Delta E = E - E_f - (E_{CO_2} \times N_{CO_2}) / N_{CO_2}$. This would remove the contribution of the CO₂ molecules, this quantity should be calculated for increasing numbers of CO₂ molecules in the pores and from that a more accurate adsorption energy would be calculated.

In light of the reviewer's suggestions, we modified our definition of ΔE (we renamed it to ΔE_{CO_2} in the revised manuscript), the averaged adsorption energy of each CO₂ molecule:

$$\Delta E_{CO_2} = - \frac{E - E_f - N_{CO_2} \times E_{CO_2}}{N_{CO_2}}$$

where E, E_f, E_{CO₂}, and N_{CO₂} are the total energy of the system, the framework energy, the energy of a single CO₂ molecule, and the number of CO₂ molecules, respectively.

New results still support the original conclusion we made:

"Before phase transition, the CO₂ adsorption energy in pore B is only 6.9 kJ·mol⁻¹ per molecule larger than that in the channel between pores A and B. Such a small energy difference suggests that CO₂ molecules are likely to stay in pore B initially but can easily move into the channel under thermal excitation. After phase transition, the CO₂ adsorption

energy in both A and B pores is $14.1 \text{ kJ}\cdot\text{mol}^{-1}$ per molecule larger than that in pore A. This suggests that CO₂ molecules tend to first fill pore B before going into pore A. The combined picture of these results is that in the adsorption process: CO₂ molecules migrate from pore B to pore A via the opened channel. This is in line with the conclusions we derived from experimental results.”

We have modified the main text using new data.

The author should also address that these calculations are carried out at 0 K, and that temperature may have an effect. A more statistically sound way of calculating adsorption energies (with temperature included) is using grand canonical Monte Carlo simulations (S. Jawahery et al. *Nature Commun.* 2017, 8, 13845; J. Hartlieb et al, *J. Am. Chem. Soc.*, 2016, 138 (7), 2292–2301). As it is a statistical mechanics method, a distribution of energies and orientations are calculated. In addition, the method can be used to confirm the uptake of CO₂ in ZIF-7.

We agree with the reviewer that GCMC simulations allow us to have a complete thermodynamic description of the CO₂ interactions with ZIF-7 under experimental conditions. Thus we performed GCMC simulations and related contents are highlighted in the revised manuscript and SI. Simulated adsorption isotherms are found to have a good match with the experimental data, which confirms the uptake of CO₂ in ZIF-7. These results also confirm the initial CO₂ adsorption in pore B of ZIF-7-II. We compared the heat of adsorption of CO₂ in ZIF-7-I and ZIF-7-II with the energy difference between the two phases. Results underpin that initial adsorption in pore B of ZIF-7-II is needed to compensate for the energy penalty of the structural transition.

REVIEWERS' COMMENTS:

Reviewer #1 (Remarks to the Author):

At first, I want to wish the authors a happy new year and hope to bring some positive news. I want to thank them for extensively addressing all questions raised throughout the revision by myself and the other referees. I find the additional computational analysis of pore size distributions and modelling of adsorption isotherms and enthalpies by GCMC methods very interesting and supportive to the experimental claims. In addition, I acknowledge the efforts in restructuring the introduction and including additional definitions and find this to greatly improve the quality of the manuscript. I do however want to point out some minor points that the authors may consider: I am still a bit opposed by the term non-uniform porous structure but I acknowledge the additional definition given in the introduction and will not fight over this terminology. The author may keep it under the given definitions.

In line 63 the authors state: "...that this solves a fundamentally important issue in the understanding of structural transitions". I am not sure if this study "solves" this issue but I am sure it addresses it very well. The authors may change the phrasing here.

In line 158: 195 K is the sublimation point of CO₂ at 101 kPa, not the boiling point.

In fig 2 A on the x-axis Theta should be in italic font as in B and C the Q should be italic as well. The authors nicely use the adsorption enthalpies to estimate the energetic driving force for the transition and Fig 4 is a great addition to the manuscript. Well done. I believe this additional computational effort underlines the experimental results and now provides a full picture of the phase transition in ZIF-7 upon CO₂ adsorption.

Over all, I think this work presents a great addition to the growing field of stimuli-responsive porous solids and it has been a pleasure revising the manuscript. The authors have performed a detailed revision of the manuscript and addressed all points raised by myself and the other referee. I therefore recommend publication in Nature Communication while considering the minor points raised above.

I also do not declare any conflicts with the addition of two authors from the initial submission as their contributions are clearly documented in the author contribution section.

Reviewer #2 (Remarks to the Author):

the authors have done an extensive job to address each reviewers' questions, and overall the authors did successfully address the CO₂ sorption mechanism into this flexible ZIF-7 well for the first time.

Reviewer #3 (Remarks to the Author):

It is clear from reading the author responses to review comments, the revised manuscript and supplementary information that the authors have invested a lot of effort into the revision of the manuscript.

I am satisfied with the response to earlier comments, and appreciate the addition of GCMC simulations to the manuscript in particular. This indeed helps to explain adsorption site energies, as well as confirming the adsorbed amount of CO₂.

The addition of the sample input file for DFT calculations is valued, and will also help the reproducibility of the work, it would also be preferred to also include sample input files for the GCMC simulations.

Response to Reviewers' Comments

We are delighted to know that the reviewers are satisfied with the revision we made previously. Here is our point-to-point response to the remaining concerns raised by the reviewers.

Reviewer #1:

At first, I want to wish the authors a happy new year and hope to bring some positive news. I want to thank them for extensively addressing all questions raised throughout the revision by myself and the other referees. I find the additional computational analysis of pore size distributions and modelling of adsorption isotherms and enthalpies by GCMC methods very interesting and supportive to the experimental claims. In addition, I acknowledge the efforts in restructuring the introduction and including additional definitions and find this to greatly improve the quality of the manuscript. I do however want to point out some minor points that the authors may consider:

1. I am still a bit opposed by the term non-uniform porous structure but I acknowledge the additional definition given in the introduction and will not fight over this terminology. The author may keep it under the given definitions.

Thanks to the reviewer for appreciating our efforts in improving the manuscript. In light of the above comment, we keep the term “non-uniform porous structure” under the definitions given in the current version.

2. In line 63 the authors state: “...that this solves a fundamentally important issue in the understanding of structural transitions”. I am not sure if this study “solves” this issue but I am sure it addresses it very well. The authors may change the phrasing here.

We thank the reviewer for pointing out the misleading phrase we used. According to the reviewer’s suggestion, we have rephrased the sentence to “...that this well addresses a fundamentally important issue...”

3. In line 158: 195 K is the sublimation point of CO₂ at 101 kPa, not the boiling point.

We have corrected our statement by changing “the boiling point of CO₂” to “the sublimation point of CO₂ at 101 kPa”.

4. In fig 2 A on the x-axis Theta should be in italic font as in B and C the Q should be italic as well.

Thanks to the reviewer for urging us to be more professional in making figures. We have changed the x-axis Theta of Fig 2 A and the Q of Fig 2 B & C to be in italic font.

The authors nicely use the adsorption enthalpies to estimate the energetic driving force for the transition and Fig 4 is a great addition to the manuscript. Well done. I believe this additional computational effort underlines the experimental results and now provides a full picture of the phase transition in ZIF-7 upon CO₂ adsorption.

Over all, I think this work presents a great addition to the growing field of stimuli-responsive porous solids and it has been a pleasure revising the manuscript. The authors have

performed a detailed revision of the manuscript and addressed all points raised by myself and the other referee. I therefore recommend publication in Nature Communication while considering the minor points raised above.

I also do not declare any conflicts with the addition of two authors from the initial submission as their contributions are clearly documented in the author contribution section.

Reviewer #2:

The authors have done an extensive job to address each reviewer's questions, and overall the authors did successfully address the CO₂ sorption mechanism into this flexible ZIF-7 well for the first time.

Thanks to the reviewer for recognizing the novelty of our work as well as our efforts in improving the manuscript.

Reviewer #3:

It is clear from reading the author responses to review comments, the revised manuscript and supplementary information that the authors have invested a lot of effort into the revision of the manuscript.

I am satisfied with the response to earlier comments, and appreciate the addition of GCMC simulations to the manuscript in particular. This indeed helps to explain adsorption site energies, as well as confirming the adsorbed amount of CO₂.

The addition of the sample input file for DFT calculations is valued, and will also help the reproducibility of the work; it would also be preferred to also include sample input files for the GCMC simulations.

Thanks to the reviewer for appreciating our efforts in improving the manuscript. As requested, we have included sample input files for the GCMC simulations as supplementary files.